# Rare coding variants in *NOX4* link high ROS levels to psoriatic arthritis mutilans

Sailan Wang [iD][1], Pernilla Nikamo[1], Leena Laasonen[2], Bjorn Gudbjornsson [iD][3], Leif Ejstrup[4], Lars Iversen [iD][5], Ulla Lindqvist[6], Jessica J Alm [iD][7], Jesper Eisfeldt[8,9], Xiaowei Zheng[9], Sergiu-Bogdan Catrina [iD][9,10], Fulya Taylan [iD][8,9], Raquel Vaz[9], Mona Ståhle [iD][1,11,12] & Isabel Tapia-Paez [iD][1,12]✉

## Abstract

**Psoriatic arthritis mutilans (PAM) is the rarest and most severe form of psoriatic arthritis, characterized by erosions of the small joints and osteolysis leading to joint disruption. Despite its severity, the underlying mechanisms are unknown, and no susceptibility genes have hitherto been identified. We aimed to investigate the genetic basis of PAM by performing massive parallel sequencing in sixty-one patients from the PAM Nordic cohort. We found rare variants in the NADPH oxidase 4 (*NOX4*) in four patients. In silico predictions show that the identified variants are potentially damaging. NOXs are the only enzymes producing reactive oxygen species (ROS). NOX4 is specifically involved in the differentiation of osteoclasts, the cells implicated in bone resorption. Functional follow-up studies using cell culture, zebrafish models, and measurement of ROS in patients uncovered that these *NOX4* variants increase ROS levels both in vitro and in vivo. We propose *NOX4* as the first candidate susceptibility gene for PAM. Our study links high levels of ROS caused by *NOX4* variants to the development of PAM, offering a potential therapeutic target.**

**Keywords** NADPH Oxidase 4 (*NOX4*); Psoriatic Arthritis Mutilans; Reactive Oxygen Species (ROS); Hydrogen Peroxide; Osteoclast Differentiation
**Subject Categories** Genetics, Gene Therapy & Genetic Disease; Musculoskeletal System; Skin

## Introduction

Psoriasis is a common inflammatory skin disease characterized by an abnormal hyperproliferation of keratinocytes, activated dendritic cells and infiltration of T lymphocytes in lesions (Lin et al, 2018). The incidence of psoriasis in Europeans is ~3%, and it has been estimated that ~30% of psoriasis patients develop psoriatic arthritis (PsA), a systemic chronic inflammatory disease with clinical features such as arthritis, enthesitis, dactylitis, tendonitis, and cutaneous psoriasis (Cafaro and McInnes, 2018). PsA is often classified into five subtypes: distal interphalangeal predominant, asymmetric oligoarticular, symmetric polyarthritis, spondylitis, and psoriatic arthritis mutilans (PAM) (Moll and Wright, 1973). PAM is the rarest and most severe form of PsA and is characterized by the shortening of one or more digits, due to severe osteolysis of the bones, a deformity known as "digital telescoping" or "opera glass finger". PAM patients suffer from severe joint destruction causing flail joints, and the progress of the deformities is rapid once the disease starts (Laasonen et al, 2020; Mochizuki et al, 2018). The skin phenotype in PAM patients is often described as mild (Gudbjornsson et al, 2013). Even though the overall prevalence of PAM is uncertain, several case studies have reported on PAM patients in different populations (Laasonen et al, 2015; Mochizuki et al, 2018; Perrotta et al, 2019; Qin and Beach, 2019), and in a Nordic PAM study it was estimated to have a prevalence of 3.7 cases per million habitants (Gudbjornsson et al, 2013). Clinical and radiographic details of patients in the Nordic PAM study have been described (Gudbjornsson et al, 2013; Laasonen et al, 2015; Laasonen et al, 2020; Lindqvist et al, 2017; Mistegard et al, 2021).

Genetic factors play an important role in the development of psoriasis and PsA, with dozens of susceptibility genes identified, and many but not all genetic signals overlapping (Cafaro and McInnes, 2018). Most of the known susceptibility genes act via the HLA locus, IFN, NF-κB, and IL23/17 signaling pathways, with some genes involved in skin barrier integrity such as *LCE3B-LCE3C* (Cafaro and McInnes, 2018). It is also thought that genetic and environmental factors such as smoking, injuries and infections play a role in the etiology of psoriatic disorders (Yan et al, 2021). Humans have suffered from PAM since ancient times, skeletal

[1]Division of Dermatology and Venereology, Department of Medicine, Solna, Karolinska Institutet, Stockholm, Sweden. [2]Helsinki Medical Imaging Center, Helsinki University Central Hospital, Helsinki, Finland. [3]Centre for Rheumatology Research, University Hospital and Faculty of Medicine, University of Iceland, Reykjavik, Iceland. [4]Department of Rheumatology, Odense University Hospital, Odense, Denmark. [5]Department of Dermatology, Aarhus University Hospital, Aarhus, Denmark. [6]Department of Medical Sciences, Rheumatology, Uppsala University, Uppsala, Sweden. [7]Department of Microbiology, Tumor and Cell Biology & National Pandemic Center, Karolinska Institutet, Stockholm, Sweden. [8]Department of Clinical Genetics, Karolinska University Hospital, Stockholm, Sweden. [9]Department of Molecular Medicine and Surgery, Karolinska Institutet, Stockholm, Sweden. [10]Center for Diabetes, Academic Specialist Center, Stockholm, Sweden. [11]Dermatology and Venereology Clinic, Karolinska University Hospital, Stockholm, Sweden. [12]These authors contributed equally: Mona Ståhle, Isabel Tapia-Paez. ✉E-mail: isabel.tapia@ki.se

remains with characteristic lesions of PAM have been found in a Byzantine monastery in Israel (Zias and Mitchell, 1996). Today, PAM has been reported in many studies from all over the world, reviewed in (Bruzzese et al, 2013), but the true prevalence of PAM is difficult to determine due to difficulties in clinical diagnosis and lack of biomarkers.

The nicotinamide adenine dinucleotide phosphate oxidase 4 (*NOX4*) gene (OMIM: 605261) encodes a protein that contains six transmembrane domains and in its cytosolic part, a flavin adenine dinucleotide (FAD) and a NADPH-binding domain. NOX4 is an enzyme involved in the production of reactive oxygen species (ROS), a group of highly reactive molecules important in the regulation of signal transduction (Brown and Griendling, 2009). NOX4 predominantly generates hydrogen peroxide ($H_2O_2$) and is constitutively active, unlike the other members of the oxidase family (Schroder, 2019). *NOX4* is expressed in many cell types, including keratinocytes and osteoclasts (Brown and Griendling, 2009). Several studies have linked high levels of ROS to conditions such as cancer, inflammatory diseases, vascular disease, diabetes and osteoporosis (Yang and Lian, 2020). Furthermore, during bone formation, the balance between osteoblasts (bone-forming cells) and osteoclasts (bone-resorbing cells) differentiation and activity is thought to be affected by ROS (Schroder, 2015). In vitro and in vivo studies have shown that increased production of NOX4 leads to increased osteoclastogenesis (Garrett et al, 1990; Lee et al, 2005; Schroder, 2019; Yang et al, 2001). Abnormal regulation of osteoclasts activity is involved in pathological bone resorption in osteoporosis, autoimmune arthritis and bone cancer (Takayanagi, 2021). A previous study identified an intronic SNP that increases the expression of *NOX4* being associated with reduced bone density and increased markers for bone turnover in middle-aged women compared to normal controls (Goettsch et al, 2013). Conversely, studies in mice show that depletion of NOX4 leads to increased trabecular bone density, and inhibition of NOX4 prevents bone loss (Yang et al, 2001).

In this study, we applied massive parallel sequencing to the whole PAM cohort and found that rare variants in the *NOX4* gene found in four PAM patients might significantly increase the levels of ROS and specifically the levels of $H_2O_2$. To test the hypothesis, we applied in vitro and in vivo models, including patient-derived osteoclasts, stable cell lines overexpressing the variants found, direct measurement of ROS in patient blood samples, and zebrafish models. Further genetic analysis of patients without *NOX4* pathogenic variants revealed other rare variants potentially pathogenic in genes related to *NOX4*. All the data obtained indicate a putative connection between high levels of ROS and the development of PAM.

## Results

### Rare *NOX4* variants in PAM patients

To investigate the pathomechanism of PAM, we looked for the presence of rare variants in patients from the Nordic PAM cohort ($n = 61$). We applied paired-end short-read sequencing to all the patients. The parameters utilized for the study design and the filtering for rare variants are outlined in Fig. 1. We found two rare variants (MAF < 0.0001) in the *NOX4* gene in two Swedish patients and one variant with low frequency (MAF < 0.001) in two additional patients from Denmark (Fig. 2A; Table 1).

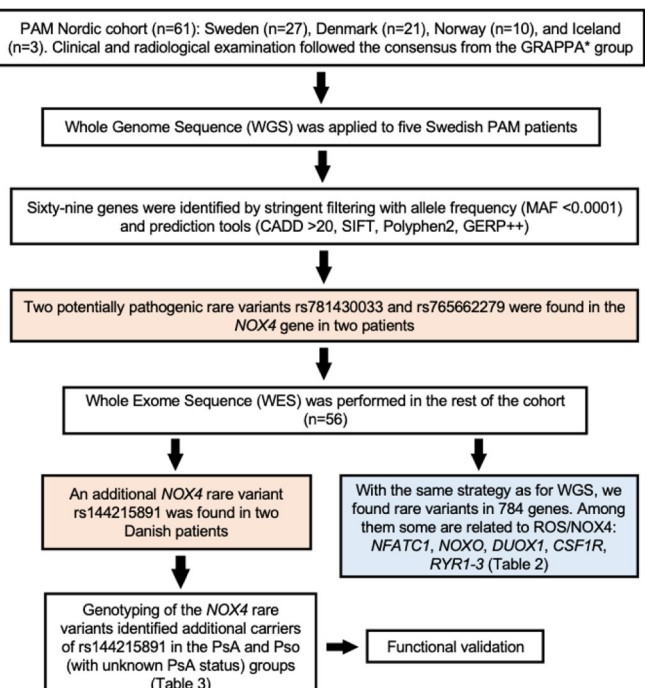

**Figure 1. Flowchart outlining the study design and the criteria applied in the filtering of rare variants found in PAM patients by next-generation sequencing.**

Data information: *GRAPPA Group for Research and Assessment of Psoriasis and Psoriatic Arthritis. Source data are available online for this figure.

One of the identified variants, rs781430033 (c.1533dup; p.Y512I*fs*X20), inserts an extra T nucleotide causing a frameshift and shortening the protein by 46 aa (Table 1). It has a frequency of 2/230742 in GnomAD and 2/120600 in ExAC databases. The second variant is a missense variant, rs765662279 (c.1105 G > T; p.V369F) (Table 1), with a frequency in the population of 9/245174 in GnomAD and 4/121010 in ExAC databases. In silico predictions of the rs765662279 with the Combined Annotation Dependent Depletion (CADD) tool in GRCh37-v1.6 resulted in a score of 23 (a CADD score of 20, meaning that the variant is in the top 1% of most deleterious substitutions in the human genome) (Rentzsch et al, 2019). The third variant—rs144215891 (c.1535 A > G; p.Y512C) found in Danish patients is present at a frequency of 77/139948 in GnomAD and 71/120620 in ExAC (Table 1). The *NOX4* variants p.Y512I*fs*X20 and p.Y512C are located one base pair apart from each other, and both affect the NADPH-binding site region. The variant p.V369F affects the FAD binding domain (Fig. 2A,D). All variants found are heterozygous and have been confirmed by Sanger sequencing of genomic DNA (Fig. 2B).

### Radiological examination of PAM patients

Radiographical examination of three PAM patients carrying rare variants in *NOX4*, PAM14 (*NOX4*^Y512IfsX20^), PAM27 (*NOX4*^V369F^), and PAM218 (*NOX4*^Y512C^), showed typical features of PAM (Fig. 2C). Patient radiographs of the hands showed shortening of

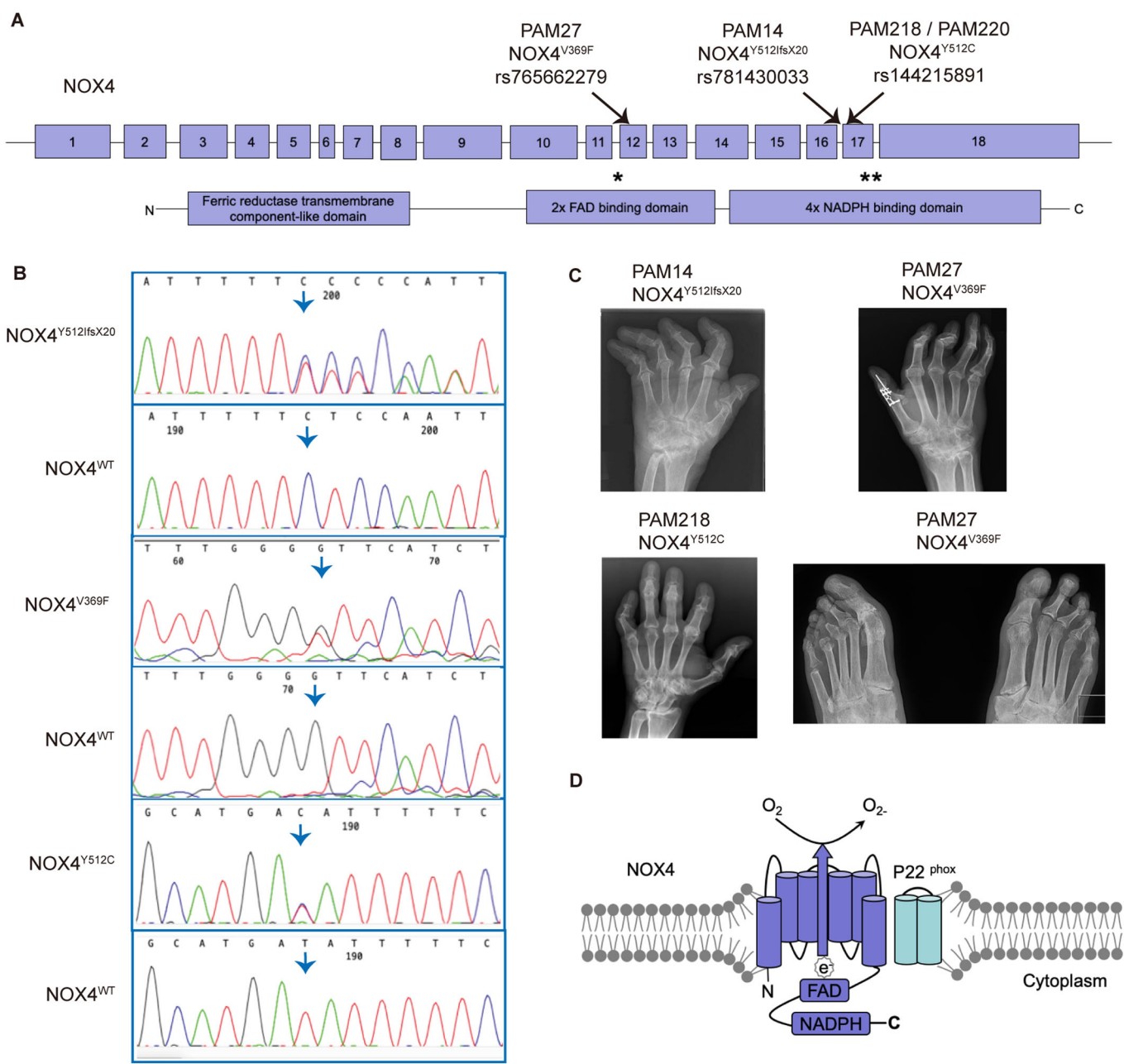

**Figure 2. Next-generation sequencing reveals rare variants in the *NOX4* gene in PAM patients.**

(A) *NOX4* gene structure is shown, exons are denoted as boxes, and the three rare variants found in PAM patients are marked by arrows; below, protein domains are indicated, stars show the position of the variants in the protein. (B) Sanger sequencing of the patients validated the findings from next-generation sequencing, heterozygous variants in *NOX4* are shown: p.V369F, p.Y512IfsX20, and p.Y512C. Arrows highlight the nucleotide change or the start of the frameshift. (C) Patient radiographs of the hands show Pencil-in-cup deformities in metacarpophalangeal joints, osteolysis and ankylosis in proximal interphalangeal joints and destruction of the wrist (os carpale) (severe in PAM14, milder in PAM218). Feet: Severe osteolysis of the interphalangeal (IP) joints on the left side. Ankylosis of the first IP joint. The mutations found in patients are denoted *NOX4^Y512IfsX20^*, *NOX4^Y512C^*, and *NOX4^V369F^*. (D) Molecular structure of NOX4. Six transmembrane domains are represented by cylinders, cytosolic domains including FAD and NADPH-binding domains are shown as boxes. e electron, FAD flavin cofactor, NADPH nicotinamide adenine dinucleotide phosphate.

the digits, severe destruction of the distal joints with pencil-in-cup deformities and osteolysis. In PAM27, severe destruction of the wrist (os carpale) was visible on the radiographs along with severe osteolysis of the interphalangeal joints in the left foot. Ankylosis of the first interphalangeal (IP) joint was also visible.

## Other rare variants related to NOX4 in PAM patients

As the *NOX4* variants were observed in a limited number of patients (4 out of 61), we extended our analysis to genes involved in the ROS/NOX4 pathway. Interestingly, we found eight additional

**Table 1. NOX4 rare coding variants in PAM patients.**

| Patient ID | Country of origin | Gender | Position in hg19 | Gene | Nucleotide change NM_016931.5 | aa change | dbSNP | Freq in GnomAD | Freq SweGene n = 2000 | Freq Danish genomes n = 150 | CADD scaled | Age | NGS |
|---|---|---|---|---|---|---|---|---|---|---|---|---|---|
| PAM14 | Swe | F | Chr11:8,906,9095 | NOX4 | c.1533dupT | p.Y512IfsX20 | rs781430033 | insT=0.00001 | 0 | 0 | N.D. | 80 | WGS |
| PAM27 | Swe | M | Chr11:89,106,630 | NOX4 | c.1105 G > T | p.V369F | rs765662279 | A = 0.00004 | 0 | 0 | 23 | 75 | WGS |
| PAM218 | Den | M | Chr11:89,069,094 | NOX4 | c.1535 A > G | p.Y512C | rs144215891 | C = 0.000558 | 0.0005 | 0 | 25 | 56 | WES |
| PAM 220 | Den | M | Chr11:89,069,094 | NOX4 | c.1535 A > G | p.Y512C | rs144215891 | C = 0.000558 | 0.0005 | 0 | 25 | 64 | WES |

*Swe Sweden, Den Denmark, M male, F female, Chr chromosome, aa amino acid, CADD Combined Annotation Dependent Depletion, NGS next-generation sequencing, WGS whole-genome sequencing, WES whole-exome sequencing. The Swedish frequency datasets are from Swegen (n = 2000) Ameur et al, 2017a; Data ref: (Ameur et al, 2017b). The Danish frequency is from publicly available Danish genomes (n = 150) (Maretty et al, 2017a); Data ref: (Maretty et al, 2017b).*

rare and potentially pathogenic variants in genes that are implicated in ROS pathways, such as *NFATC1, NOXO, DUOX1, CSF1R, RYR1, RYR2*, and *RYR3* summarized in Table 2.

The nuclear transcription factor of the activated T cells c1 (*NFATC1*) gene is a key transcription factor with an essential role in osteoclast differentiation (Takayanagi, 2021). The variant found has a high CADD score of 29.3, and the frequency in the population is very low $T = 0.000019$ (5/264690, TOPMED) and $T = 0.000141$ (17/120380, ExAC) (Table 2). Other potentially relevant rare variants were found in the ryanodine receptors *RYR1, RYR2*, and *RYR3*. The variants found in the *RYR* genes are extremely rare or non-existing in databases with CADD scores ranging from 25.8 to 33 and predicted to be damaging or deleterious by in silico tools such as PolyPhen (Adzhubei et al, 2013) or SIFT (Ng and Henikoff, 2003). *CSF1R* is the receptor of the Colony-stimulating factor-1 (CSF-1) that is released from osteoblasts and stimulates the proliferation of osteoclast progenitors (Wittrant et al, 2009). NADPH oxidase organizer 1, *NOXO*, and Dual oxidase 1, *DUOX1* are enzymes that produce ROS (Ameziane-El-Hassani et al, 2015) (Table 2). We also analyzed structural variants (SVs) in the five PAM patients sequenced by whole-genome sequencing. We used the FindSV pipeline to filter the variants. No SVs in *NOX4* nor in any other gene related to *NOX4* pathways were found (see Dataset EV1).

## Genotyping of NOX4 variants in other cohorts of psoriasis, PsA, and healthy controls

In order to elucidate if the variants found in *NOX4* are specific to PAM, we genotyped the three SNPs rs781430033 (*NOX4^Y512IfsX20^*), rs765662279 (*NOX4^V369F^*), and rs144215891 (*NOX4^Y512C^*) in previously described case-control cohorts of psoriasis ($n = 1874$) and age and gender-matched healthy controls from Sweden ($n = 484$). (Nikamo et al, 2020). We found five additional carriers of the variant *NOX4^Y512C^*, three in the PsA group and two in a group of 820 psoriasis patients with unknown PsA status (Table 3). In the psoriasis cases without arthritis, and in healthy controls, none of the variants was detected. No additional carriers of the rare variants *NOX4^Y512IfsX20^* and *NOX4^V369F^* were found in any of the groups investigated.

Next, in the same cohorts, we investigated the *NOX4* intronic variant rs11018268 (MAF $C = 0.211354$, GnomAD). A previous study by Goettsch et al (Goettsch et al, 2013) showed that the CC and CT genotypes of rs11018268 are associated with higher levels of *NOX4* expression, decreased bone density and an increased level of bone turnover markers (Goettsch et al, 2013). In our cohort, we found that two carriers of *NOX4^Y512C^*, patients PAM218 and PAM 220, are also carriers of the rs11018268-CC and rs11018268-CT alleles, respectively, and nine other PAM patients carried the CT allele. The PAM cohort was not enriched for the CC and CT alleles compared to the other groups analyzed (Appendix Table S1).

## Expression of NOX4 is higher in HEK293 cell models of NOX4^Y512IfsX20^, NOX4^V369F^ and NOX4^Y512C^

To study the functional relevance of the three *NOX4* variants found in PAM, we generated HEK293 stable transfected cell lines overexpressing *NOX4* wild-type (wt) and each of the three identified *NOX4* variants. The sequences of all the expression

**Table 2. Other rare variants affecting the generation ROS in PAM patients.**

| Patient ID | Country of origin | Gender | Position in hg19 | Gene | Nucleotide change | aa change | dbSNP | Freq in GnomAD | Freq SweGene n = 2000 | Freq Danish genomes n = 150 | CADD scaled | Age | NGS |
|---|---|---|---|---|---|---|---|---|---|---|---|---|---|
| PAM 15 | Swe | M | Chr18:77,208,845 | NFATC1 | c.1450 C > T | p.R484C | rs375389433 | T = 0.000032 | 0.0010 | 0 | 29.3 | 50 | WES |
| PAM 34 | Swe | M | Chr19:38,956,793 | RYR1 | c.2933 C > T | p.P978L | rs200124278 | T = 0.000255 | 0.0005 | 0 | 25.8 | 64 | WES |
| PAM 26 | Swe | M | Chr1:237,794,770 | RYR2 | c.6484 A > C | p.M2162L | rs1680242037 | | 0 | 0 | 26.5 | 82 | WES |
| PAM 220 | Den | M | Chr5:149,459,783 | CSF1R | c.424 C > T | p.R142C | rs147811334 | A = 0.000008 | 0 | 0 | 23.4 | 64 | WES |
| PAM 34 | Swe | M | Chr16:2,030,030 | NOXO | c.569 T > C | p.E189P | Not in db | | 0 | 0 | 23.8 | 64 | WES |
| PAM 208 | Den | F | Chr15:34,140,586 | RYR3 | c.13592 T > C | p.I4531T | Not in db | | 0 | 0 | 28.1 | 80 | WES |
| PAM 26 | Swe | M | Chr15:34,140,621 | RYR3 | c.13627 C > T | p.P4543A | rs762840378 | T = 0.000050 | 0 | 0 | 33 | 82 | WES |
| PAM 201 | Den | F | Chr15:45,427,861 | DUOX1 | c.685 C > T | p.R229W | rs1896290789 | T = 0.000007 | 0 | 0 | 16.32 | 61 | WES |

Swe Sweden, Den Denmark, M male, F female, Chr chromosome, aa amino acid, CADD Combined Annotation Dependent Depletion, NGS next-generation sequencing, WGS whole-genome sequencing, WES whole-exome sequencing.

constructs were confirmed by Sanger sequencing (Appendix Fig. S1). Stably transfected cells with $NOX4^{wt}$ and the three rare variants $NOX4^{Y512IfsX20}$, $NOX4^{V369F}$ and $NOX4^{Y512C}$, were analyzed for $NOX4$ expression by Real-Time qRT-PCR (Fig. 3A; Appendix Fig. S2). Interestingly, all three rare variants resulted in enhanced over-expression of $NOX4$ mRNA compared to the overexpression of $NOX4^{wt}$. The highest expression was observed for the $NOX4^{Y512IfsX20}$ variant (Fig. 3A). All primers used are listed in Appendix Table S2.

As it is known that NOX4 is involved in the ROS pathway and specifically generates mainly $H_2O_2$, we hypothesized that the levels of oxidant species might be affected. To evaluate the effect on ROS production in the stably transfected cell lines, we performed Electron Paramagnetic Resonance (EPR), a highly sensitive method that allows the direct detection of radicals (Dikalov et al, 2018; Suzen et al, 2017) (Fig. 3B). Furthermore, we assessed overall ROS in HEK293 cells through DCFH-DA stainings (Fig. EV1), and employed a specific probe to evaluate $H_2O_2$ levels (Fig. 3C). We observed higher ROS and $H_2O_2$ levels in all cells expressing the rare variants compared to those overexpressing $NOX4^{wt}$ (Figs. 3B,C and EV2).

## ROS levels are higher in patients PAM12 and PsA961

To validate the involvement of the reactive oxygen species (ROS) in the development of PAM, we recruited individuals with Psoriasis (n = 7), PsA (n = 6), PsA961 (PsA patient carrier of $NOX4^{Y512C}$ detected through genotyping, see Table 3) PAM (PAM12 and PAM37), as well as age and gender-matched healthy controls (n = 10). ROS levels were measured in fresh peripheral blood samples by using the sensitive EPR method. ROS expression was elevated in patients PAM12 and PsA961 compared to all other groups (Fig. 4A). The results suggest that the variant $NOX4^{Y512C}$ affecting the NADPH-binding domain is responsible for the elevated ROS production. NOX4 protein expression of PsA961 in osteoclasts is in line with this observation (Fig. S3). Analysis of PAM37 showed no difference in ROS production (Fig. 4B). One explanation for this finding may be that one day before the sample collection, the patient PAM37 was treated with etanercept, an anti-TNF-α drug. Several studies have shown that TNF-α induces the production of ROS (Kim et al, 2010; Zelova and Hosek, 2013), and the medication could have affected the level of ROS. It should be noted here that we did neither find any $NOX4$ mutations in PAM12 and PAM37 nor mutations in any other genes related to $NOX4$.

## The levels of ROS in patient PsA961 are decreased after treatment with adalimumab

The patient PsA961 was treated with adalimumab, a monoclonal antibody that suppresses tumor necrosis factor-alpha (TNFα) and inhibits ROS production (Blaser et al, 2016) followed by ixekizumab for his skin psoriasis. The patient had severe skin psoriasis since 10 years and presented a very mild PsA phenotype of recent onset with intermittent pain in his wrist and knees and in one finger. Radiographs of the hands were normal at the start of adalimumab. We studied whether the overproduction of ROS in the PsA961 patient was affected by the treatment. Fresh whole blood was obtained at four different time points: before treatment started, six months, nine months, and seventeen months while on adalimumab treatment and 5 months following treatment start of

**Table 3.** *NOX4* rare variants in PsO, PsA, and control groups.

| Marker | Alleles[a] | PsA ($n = 492$) | PSO No PsA ($n = 562$) | PSO unknown PsA ($n = 820$) | PAM ($n = 63$) | Controls ($n = 484$) |
|---|---|---|---|---|---|---|
| rs781430033 | A/T | 478/0/0 | 552/0/0 | 815/0/0 | 62/1/0 | 478/0/0 |
| rs765662279 | C/T | 479/0/0 | 552/0/0 | 818/0/0 | 62/1/0 | 480/0/0 |
| rs144215891 | T/C | 478/3[b]/0 | 550/0/0 | 816/2/0 | 61/2/0 | 480/0/0 |

[a]Major/minor.
[b]PsA patients carrying rs144215891 are: PsA961, PsA252, and PsA690.

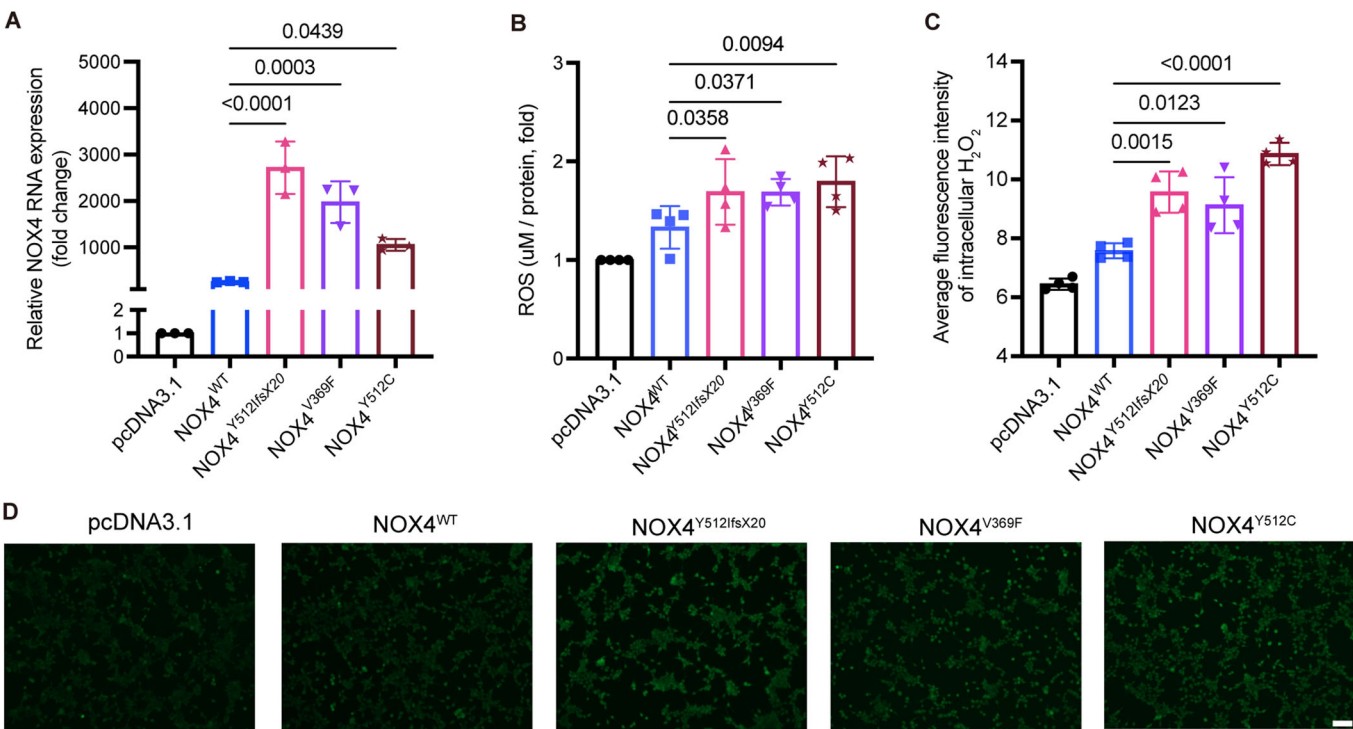

**Figure 3.   HEK293 cell lines overexpressing the rare variants found in *NOX4* increased *NOX4* expression and reactive oxygen species (ROS) generation compared to cells overexpressing *NOX4^wt^*.**

(**A**) The expression of *NOX4* was analyzed by qRT-PCR of cells extracted from HEK293 stable transfected cells expressing pcDNA3.1 (empty vector), *NOX4^wt^* (wild-type), *NOX4^Y512IfsX20^*, *NOX4^Y512C^*, and *NOX4^V369F^*. β-*actin* levels were used as a loading control. Relative levels of *NOX4* were quantified from three independent experiments. $N = 3$. (**B**) All *NOX4* rare variants expressing cells have a higher generation of ROS compared to *NOX4^wt^*. $N = 4$. (**C**) The generation of $H_2O_2$ was assessed using the 1 mM BioTracker Green $H_2O_2$ live-cell dye for 20 min in HEK293 stable transfected cells. The mean fluorescence intensity of $H_2O_2$ was quantified using ImageJ. $N = 4$. (**D**) Representative immunofluorescence images of $H_2O_2$ levels in HEK293 cells expressing NOX4 rare variants. Scale bar: 100 μm. Data information: $H_2O_2$ hydrogen peroxide. $N =$ biological replicates. For graphs (**A–C**), data are shown as mean ± SEM based on the ordinary one-way ANOVA compared to *NOX4^wt^*. Source data are available online for this figure.

ixekizumab (anti-IL17 agent). At all time points after biological therapy, ROS levels were normalized reaching ROS levels of healthy controls and other PsA patients without *NOX4* variants (Fig. 4C).

## PAM12-derived osteoclasts show increased differentiation and higher generation of ROS compared to cells from a healthy control

NOX4 is induced during osteoclast differentiation, the cells responsible for bone resorption (Goettsch et al, 2013). To determine if the process of osteoclastogenesis is affected in PAM, we performed in vitro osteoclast differentiation of patient-derived cells from patient PAM12 and an age- and gender-matched healthy control (C12). Osteoclast differentiation was induced in

CD14+ peripheral blood mononuclear cells by treatment with cytokines M-CSF and RANKL (Fig. 5A). We observed a higher number of differentiated osteoclasts in the PAM12 patient compared to the healthy control, determined by the presence of multinucleated TRAP-positive cells, at both day 8 and 12 of culture (Fig. 5B,C). In addition, NOX4 protein levels were higher in the PAM12-derived osteoclasts compared to C12-derived osteoclasts (Fig. 5D). To examine cellular ROS production in osteoclasts, we used 2'-7'dichlorofluorescin diacetate (DCFH-DA). DCFH-DA is a cell-permeable compound that is deacetylated by cellular esterases and oxidized by ROS into 2'-7'dichlorofluorescin (DCF). The DCF-emitted fluorescence can then be quantified by fluorescence microscopy (Eruslanov and Kusmartsev, 2010). ROS production in PAM12 osteoclasts at both days 8 and 12 of

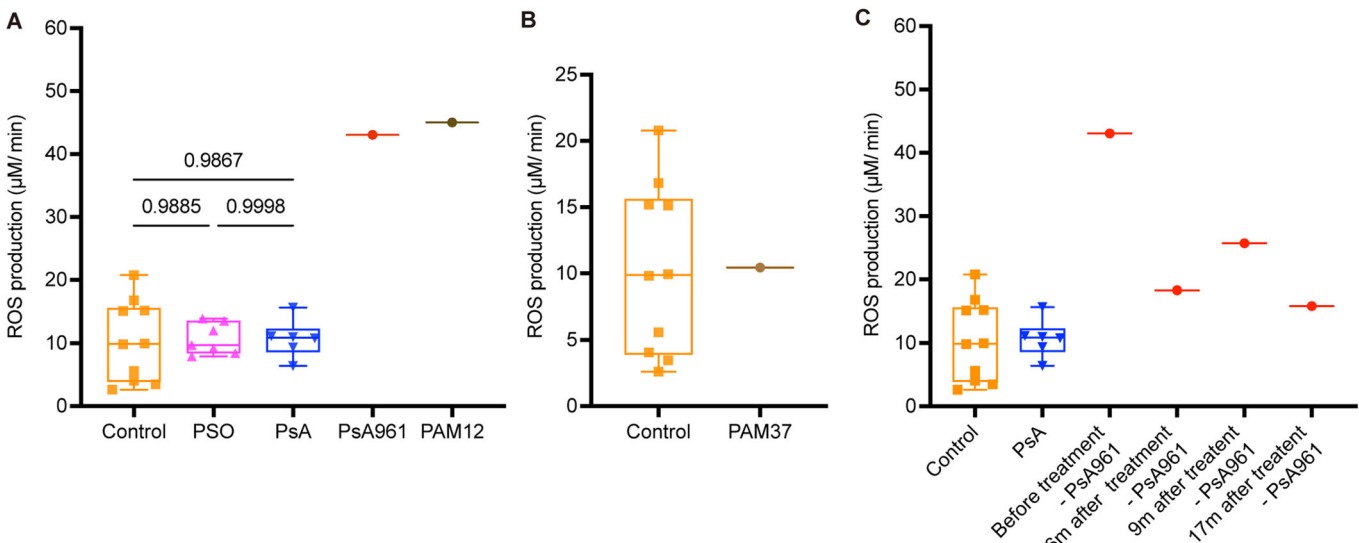

**Figure 4. Electron paramagnetic resonance (EPR) shows increased levels of ROS in the patients PAM12 and PsA961 and the impact of anti-TNF-α treatment in PsA961.**

(A) ROS measurement by EPR shows increase in PsA961 ($NOX4^{Y512C}$ carrier, red dot) and PAM12 (dark brown dot) compared to controls ($n = 10$, square), PSO ($n = 7$, triangle), PsA ($n = 6$, inverted triangle) from peripheral blood. (B) ROS measurement of the PAM37 patient (light brown dot) undergoing anti-TNFα treatment shows no difference compared to samples from healthy controls ($n = 10$). (C) ROS levels of the PsA961 patient decreased after anti-TNF treatment. Peripheral blood samples were obtained on four occasions: before treatment, six months (6 m), nine months (9 m) and seventeen months (17 m) after treatment. Data information: PsA961 is a carrier of $NOX4^{Y512C}$. $n$ = biological replicates. Each symbol represents the average measurement of two technical replicates for each individual. The box and whisker plots display data distribution through the minimum, first quartile, median, third quartile, and maximum. For graph (A), the $P$ values were calculated by one-way ANOVA, followed by Tukey's multiple comparisons correction. Source data are available online for this figure.

culture was significantly higher compared to C12 osteoclasts (Fig. 5E,F).

### PsA961- ($NOX4^{Y512C}$) derived osteoclasts show increased differentiation and higher generation of $H_2O_2$ compared to cells from a healthy control

To investigate whether $H_2O_2$ levels are altered during osteoclastogenesis in individuals carrying the $NOX4$ variant, we performed in vitro osteoclast differentiation of patient-derived cells in PsA961 (carrier of the rare variant $NOX4^{Y512C}$). For comparison, we included a healthy control and a PsA patient without any of the three rare variants (PsA77). We observed an increased number of differentiated osteoclasts in the PsA961 determined by the presence of multinucleated TRAP-positive cells, at both day 7 and day 11 (Fig. 6A). To examine intracellular $H_2O_2$ production in osteoclasts, we used the BioTracker Green $H_2O_2$ live-cell dye. Remarkably, $H_2O_2$ levels in PsA961 were significantly higher compared to PsA77 and the C17 healthy control cells on days 7, 9, and 11 (Fig. 6B,C). Furthermore, PsA961-differentiated osteoclasts exhibited elevated ROS production, as indicated by DCFH-DA, in comparison to C17 and PsA77 osteoclasts (Fig. EV2). In addition, the protein levels of NOX4 were higher in the PsA961 derived osteoclasts compared to C11-derived osteoclasts (Fig. 6D).

### $NOX4$ rare variants $NOX4^{Y512IfsX20}$, $NOX4^{V369F}$, and $NOX4^{Y512C}$ increase generation of ROS in zebrafish

To further examine the effect of the three variants found in PAM patients in vivo, we tested the ROS production in zebrafish embryos injected with $NOX4$ mRNA coding for the wild-type as well as the three rare variants (Fig. 7). We co-injected the mRNAs with a nox4 translation-blocking antisense oligonucleotide, or morpholino (nox4 atg MO), to reduce the amount of the endogenous Nox4 production and better resemble the expression patterns in patients. Protein quantification confirmed a 50% reduction in Nox4 following the injection of nox4 atg MO (Fig. 7A). Quantification of ROS by DCFH-DA revealed a higher amount of ROS production in embryos overexpressing the $NOX4$ variants compared to those overexpressing $NOX4^{wt}$ (Fig. 7C). Quantification of the fluorescence intensity in a $100 \times 100$ μm region of the embryos' trunk region (Fig. 7B) showed a significant increase for all three constructs $NOX4^{Y512IfsX20}$, $NOX4^{V369F}$, and $NOX4^{Y512C}$ (Fig. 7C,D). These results in zebrafish embryos are consistent with the results obtained from HEK293T cells and the patient's cells (Fig. 3A).

## Discussion

The NOX gene family is comprised of seven members $NOX1$-$NOX5$, $DUOX1$, and $DUOX2$ (Wegner and Haudenschild, 2020). They are specialized ROS producers and differ in their cellular and tissue-specific distributions. Impairment in the regulation of NOX expression results in pathologies such as atherosclerosis, hypertension, diabetic nephropathy, lung fibrosis, cancers, and neurodegenerative diseases (Vermot et al, 2021). $NOX4$ specifically has been linked to osteoporosis, inflammatory arthritis and osteoarthritis (Wegner and Haudenschild, 2020). The role of $NOX4$ in those pathologies has been suggested by tissue-specific-expression studies, functional biochemical assays, and observations in animal

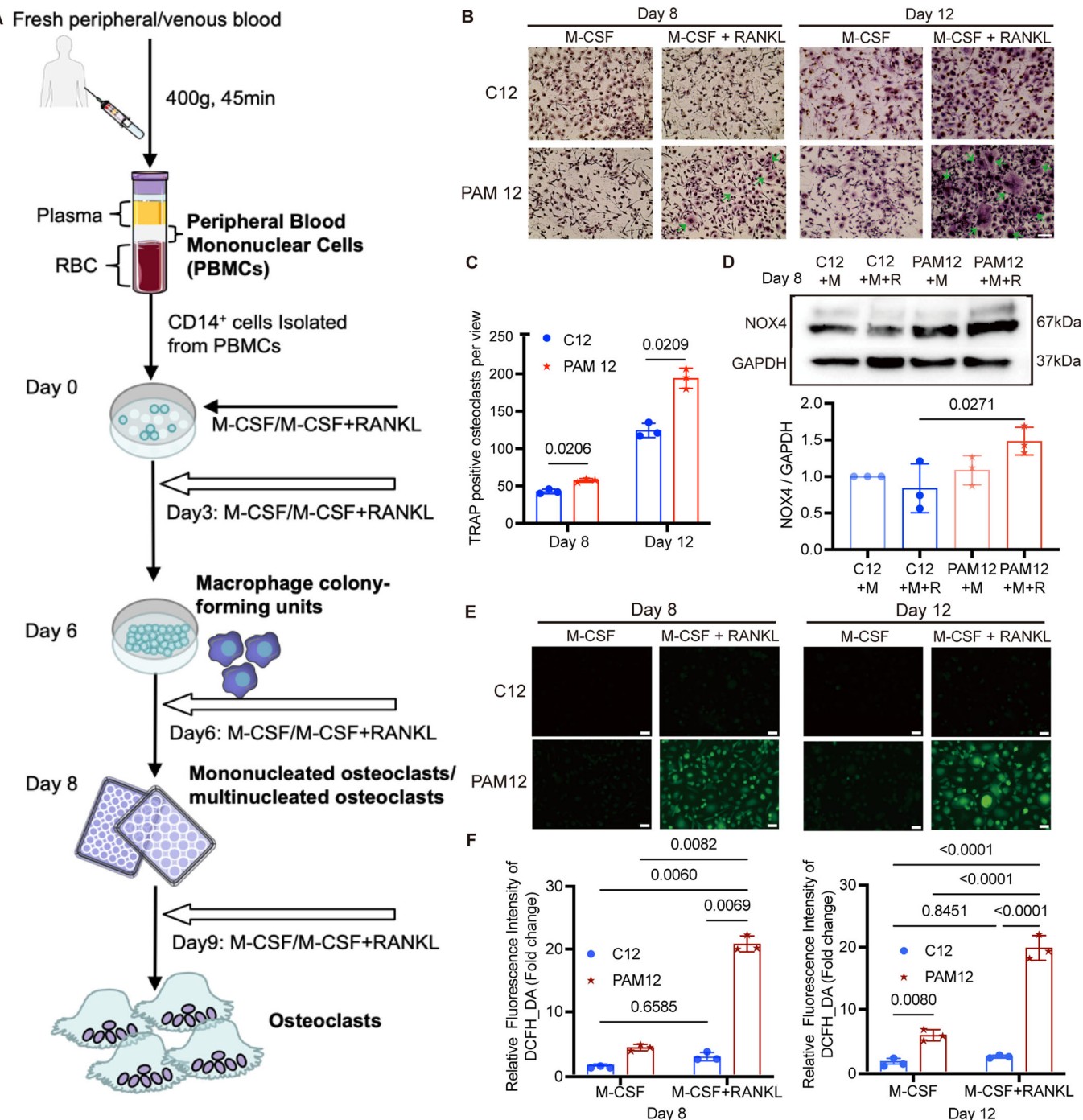

**Figure 5.** **PAM12-derived osteoclasts show enhanced differentiation and increased ROS generation activity compared to osteoclast-derived cells from healthy control.**

(A) Schematic illustration of isolation of osteoclasts from peripheral blood. More details are shown under "Methods". (B) Osteoclasts were differentiated with colony-stimulating factor (M-CSF/M) and receptor activator of nuclear factor κB ligand (RANKL/R). Tartrate-resistant acid phosphatase (TRAP) staining (violet-labeled) was used to mark differentiated osteoclast (> 3 nuclei). PAM12-derived cells show a higher number of differentiated osteoclasts (marked by arrows) compared to cells derived from a healthy control (C12). (C) The number of TRAP-positive osteoclasts per view was counted blindly by 2 persons. $N = 3$. (D) Western blot detected the NOX4 protein expression in differentiated osteoclasts derived from PAM12 and C12 on Day 8, and GAPDH was used as a loading control. $N = 3$. (E) ROS probed by DCFH-DA in cells from PAM12 was significantly higher compared to control cells at both 8 and 12 days after differentiation with M-CSF and RANKL. Representative images are shown. $N = 3$. (F) The average fluorescence intensity was quantified by ImageJ software at two time points. $N = 3$. Data information: scale bars = 100 μm. $N =$ biological replicates. (C, D, F) Error bars in figures represent mean ± SEM. One-way ANOVA with Tukey's multiple comparisons test was used to determine significant differences. Source data are available online for this figure.

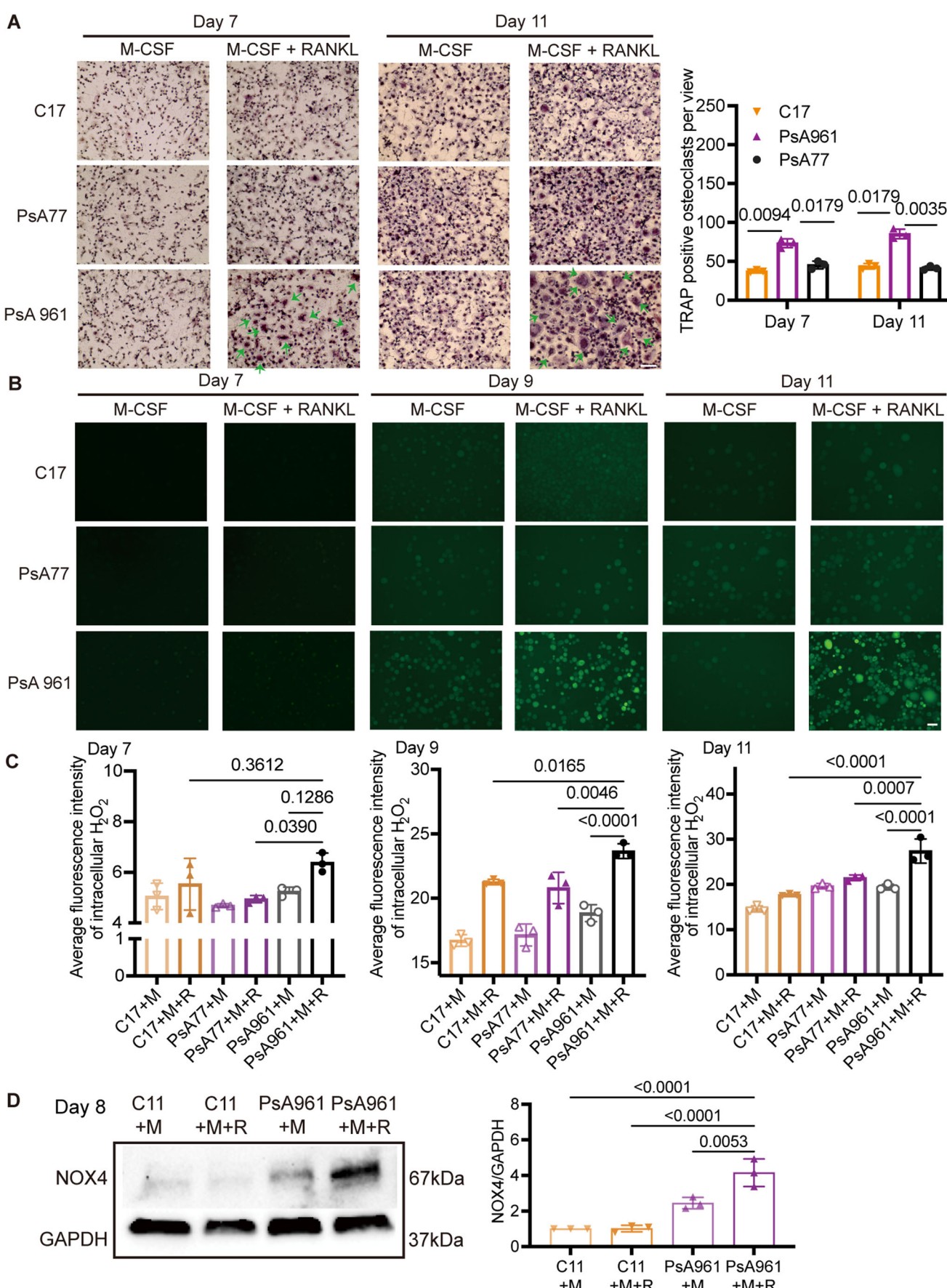

◀ **Figure 6. PsA961 (*NOX4^Y512C*) derived osteoclasts show enhanced differentiation and increased hydrogen peroxide generation activity compared to a PsA patient (non-carrier of *NOX4* rare variants) and to a healthy control.**

(**A**) Osteoclasts were induced through differentiation using colony-stimulating factor (M-CSF/M) and receptor activator of nuclear factor κB ligand (RANKL/R). Differentiated osteoclast with more than three nuclei were identified through tartrate-resistant acid phosphatase (TRAP) staining (violet-labeled). Notably, cells derived from PsA961 patient carrying the *NOX4^Y512C* variant shown a higher numbers of differentiated osteoclasts (indicated by arrows) compared to age-gender-matched PsA77 patient without *NOX4* variants. $N = 3$. (**B**) Hydrogen peroxide ($H_2O_2$) levels, probed by a green dye in cells from PsA961 patients, were significantly higher compared to control cells (C17 and PsA77) at different time points after differentiation with M-CSF and RANKL. Representative images are shown. (**C**) The average fluorescence intensity was quantified by ImageJ software at three time points. $N = 3$. (**D**) In differentiated osteoclast (Day 8) NOX4 protein levels were higher in PsA961 compared to a healthy control—C11. GADPH levels were used as a loading control. $N = 3$. Data information: $H_2O_2$ hydrogen peroxide. Scale bars = 100 μm. $N =$ biological replicates. For graphs (**A**, **C**, **D**), error bars in figures represent mean ± SEM. One-way ANOVA with Tukey's correction for multiple testing was used. Source data are available online for this figure.

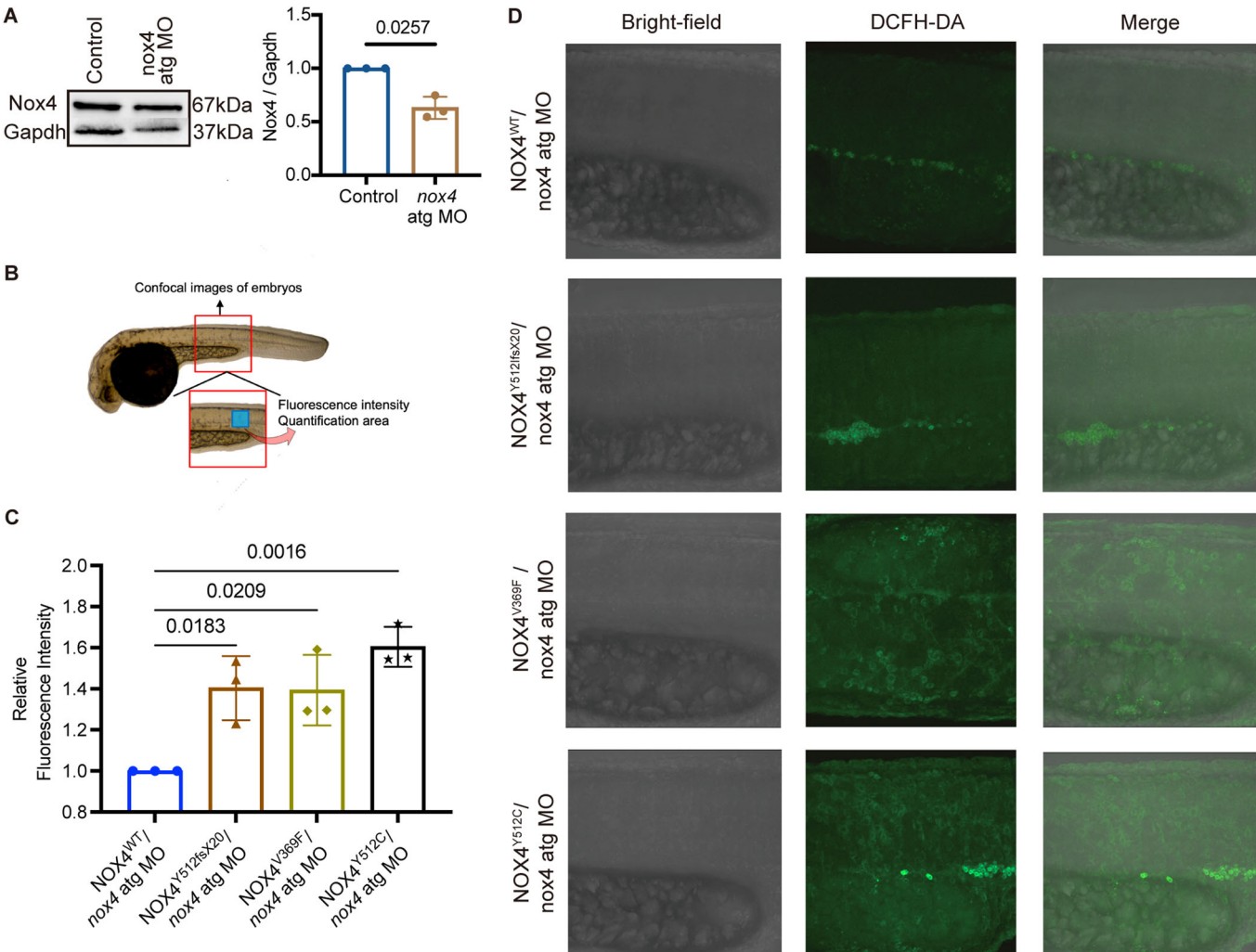

**Figure 7. ROS production is increased in zebrafish embryos injected with *NOX4* mRNA variants compared to *NOX4^wt*.**

(**A**) Endogenous Nox4 was reduced by co-injecting embryos with a translation-blocking morpholino (nox4 atg MO), resulting in a 50% reduction of Nox4 compared to mock-injected controls. Gapdh was used as loading control. $N = 3$. (**B**) ROS production was assessed by imaging zebrafish embryos (red box) and quantifying the fluorescence intensity in the area marked by the blue box. (**C**) Quantification of fluorescence intensity shows a significant increase in the embryos co-injected with *NOX4* variants mRNA and nox4 atg MO compared to *NOX4^wt* and nox4 atg MO The number of zebrafish embryos imaged in each experimental group is listed in Appendix Table S7. In total, 9–16 zebrafish embryos were used for each repeat. $N = 3$. (**D**) Representative images of injected embryos exposed to DCFH-DA (scale bars = 100 μm). Data information: DCFH-DA 2′,7′-dichlorofluorescein diacetate. $N =$ biological replicates. Values are mean ± SEM. For graph (**A**), the difference between two groups was determined by the Student's *t* test. For graph (**C**), ordinary two-way ANOVA with Tukey's multiple comparisons test was used to determine significant differences. Source data are available online for this figure.

models (Drevet et al, 2018; Goettsch et al, 2013; Gray et al, 2019), but to our knowledge no disease-causing variants in *NOX4* have yet been found in patients.

Our study identified *NOX4* as the first candidate susceptibility gene for psoriatic arthritis mutilans (PAM), the rarest and most severe form of psoriatic arthritis. Despite the severity of the disease, no specific treatment or biomarkers have been identified to date. Here, we describe three protein-coding rare variants; two missense ($NOX4^{V369F}$ and $NOX4^{Y512C}$) and one frameshift ($NOX4^{Y512IfsX20}$), in four PAM patients, all located in the cytosolic part of NOX4, affecting the FAD and NADPH-binding domains, important for the transfer of electrons and formation of ROS (Magnani et al, 2017). In our genetic analysis, we did not find any other *NOX4* rare variants in the rest of the PAM cohort ($n = 57$), but we cannot exclude that these patients may carry other variants located in intergenic or intragenic regions which would escape our analysis as most of the patients were sequenced only for exomes. It is also possible that other non-ROS-related pathways could be implicated in the development of PAM. Through further genetic analysis of the three rare variants in psoriasis, PsA and control cohorts, we found three additional carriers of the $NOX4^{Y512C}$, all in the PsA group whereas none in the psoriasis group nor in healthy controls (Table 3). Of the non-PAM patients carrying *NOX4* mutations, only PsA961 was available for further clinical examination. He presented a mild PsA phenotype affecting peripheral joints without any evidence of bone destruction. It should be noted that his PsA is of short duration, and it was his cutaneous psoriasis that motivated anti-TNF therapy. Thus, we cannot know whether he would have developed a more advanced phenotype without systemic therapeutic intervention. Also, the pathogenetic architecture in PAM is likely complex and we do not have a complete picture.

Further analysis of PAM patient sequences revealed rare variants in other genes potentially altering the levels of ROS and/ or affecting osteoclast differentiation including the transcription factor *NFATc1, CSF1R, NOXO, DUOX1, RYR1, RYR2,* and *RYR3.* Interestingly, the *NFATc1* gene is a master regulator of RANKL-induced osteoclastogenesis and the *Nfatc1* conditional knockout mouse develops osteopetrosis, a condition characterized by increased bone density due to decreased or absent osteoclast activity (Aliprantis et al, 2008). *CSF1R* is involved in osteoclast proliferation, and its suppression has been shown to attenuate pathological bone resorption in inflammatory arthritis, inflammatory bone destruction, and osteoporosis (Mun et al, 2020). *DUOX1* is part of the NADPH family and, like *NOX4*, also produces $H_2O_2$. Duox1 forms heterodimers with dual oxidase maturation factor-1 (Duoxa1), which was recently shown to be involved in osteoclast differentiation and ROS production in bone (Cheon et al, 2020). Ryanodine receptors (RyRs) are calcium ($Ca^{2+}$) channels that are responsible for $Ca^{2+}$ release from the sarcoplasmic reticulum (Liu et al, 2017). In cancer-associated bone metastasis in a mouse model, upregulation of *Nox4* results in elevated oxidization of skeletal muscle proteins, including RyR1 (Waning et al, 2015). Also, *NOXO* is involved in ROS formation, and shown to play role in angiogenesis (Brandes et al, 2016). Altogether, the variants affecting NOX4/ROS levels pathways are found in ~20% of the PAM patients.

It is interesting to note that the *NOX4* intronic SNP rs11018628, previously linked to reduced bone density and elevated plasma markers for bone turnover (Goettsch et al, 2013) is found in the

three PAM patients carrying NOX4 missense rare variants ($NOX4^{Y512C}$, $NOX4^{V369F}$), but not in the PAM patient carrying the frameshift variant ($NOX4^{Y512IfsX20}$). We also found the SNP rs11018628 in the PsA961 carrier of $NOX4^{Y512C}$. Perhaps, there could be an additive or synergistic effect of these variants on *NOX4* expression at the transcriptional level, leading to increased generation of ROS. Our analysis of the ROS levels in patient PsA961 using electron paramagnetic resonance (EPR) showed significantly increased ROS levels compared to other individuals in PsA, psoriasis and healthy controls groups (Fig. 4A). The ROS levels were similar to the levels observed in the PAM12 patient. Unfortunately, we were not able to recruit any of the other patients for the measurement.

Several factors are involved in the regulation of *NOX4*, including NF-κB, TGF-β, TNF-α, endoplasmic reticulum (ER) stress, hypoxia, and ischemia but the underlying mechanisms behind the regulation are not fully understood (Lou et al, 2018). Our results are in line with the previous observations showing that upregulation of NOX4 is linked to several pathogenic conditions, such as idiopathic pulmonary fibrosis (Amara et al, 2010), chronic obstructive pulmonary disease (Hollins et al, 2016) several cardiovascular conditions (Chen et al, 2012) and osteoporosis (Goettsch et al, 2013). In addition, *NOX4* is important for osteoclast differentiation. Osteoclasts are multinucleated cells derived from the monocyte-macrophage lineage, important for bone remodeling. They resorb the bone and its hyperactivated function is implicated in diseases such as osteoporosis, periprosthetic osteolysis, Paget's bone disease, and rheumatoid arthritis (Bi et al, 2017). Here, we evaluated osteoclast differentiation and generation of ROS in one PAM patient (PAM12) and one PsA patient carrying a *NOX4* rare variant (PsA961 carrier of $NOX4^{Y512C}$) and found that the patients showed a similar pattern compared to a PsA individual without *NOX4* rare variants and two healthy controls.

In summary, we here present novel genetic findings, supported by several lines of functional evidence for the involvement of ROS in the etiology of PAM: (i) using stably transfected HEK293 cells, we show that the rare variants result in elevated *NOX4* transcript expression and ROS generation (Fig. 3A–C), (ii) measurement of ROS in patient PAM12 (a patient without identified *NOX4* mutations) and patient PsA961 (carrier of $NOX4^{Y512C}$) showed a significant increase of ROS compared to control, psoriasis and PsA samples by EPR (Fig. 4A,C), (iii) patient-derived cells from PAM12 showed increased osteoclast differentiation with increased ROS activity compared to cells from a healthy control (Fig. 5), (iv) patient-derived cells from PsA961 (carrier of $NOX4^{Y512C}$) showed increased osteoclast differentiation and increased $H_2O_2$ activity compared to cells from a PsA individual without *NOX4* rare variants and healthy control (Fig. 6), and finally (v) using a zebrafish model, we show in vivo that the generation of ROS is significantly enhanced by all three *NOX4* rare variants found in PAM patients (Fig. 7C,D).

A limitation of this study is the lack of access to fresh blood samples from the majority of PAM patients, needed for the measurement of ROS by EPR and for osteoclast differentiation from CD14+ monocytes. Our study would have benefited from exploring ROS levels in more patients. Nevertheless, we had the possibility of testing a couple of PAM patients and a PsA (carrier of $NOX4^{Y512C}$) by EPR as a proof of concept that the generation of ROS

is indeed affected in both patients. Another consideration is that the rare variants found in *NOX4* are observed in just a few PAM patients (4 out of 61). Additional genetic analysis indicates potentially pathogenic variants in other genes found in PAM patients also affecting osteoclast differentiation and activity. Further functional validation experiments are required to test the pathogenicity of those variants.

Interestingly, the patients at risk of developing PAM may benefit from existing biological therapies applied in moderate and severe psoriasis which may reduce the generation of ROS. Another commonly used drug for treating psoriasis, methotrexate, inhibits osteoclast differentiation by inhibiting RANKL (Kanagawa et al, 2016). With the advent of effective therapies for psoriasis and psoriatic arthritis, PAM has become increasingly rare, still early diagnosis is important to avoid irreversible damage.

This study reveals a direct link to *NOX4* and ROS/H$_2$O$_2$ production in PAM pathology and gives the first strong indication of where to search for specific disease identifiers in this destructive disease. Would early intervention with existing biologic treatments be sufficient or is precision therapy essential? The disease process can be rapid in PAM resulting in irreparable damage. Early identification of those at risk and initiation of effective therapy would constitute a game changer.

# Methods

### Human samples

In this study, genomic DNA was isolated from peripheral blood mononuclear cells (PBMCs) from the Nordic PAM patient's cohort of 61 well-characterized patients previously described (Gudbjornsson et al, 2013; Laasonen et al, 2015; Laasonen et al, 2020; Lindqvist et al, 2017; Mistegard et al, 2021; Nikamo et al, 2020). The cohort consists of patients from Sweden ($n = 27$), Denmark ($n = 21$), Norway ($n = 10$), and Iceland ($n = 3$). The patients' clinical and radiographic presentations follow the consensus from the Group for Research and Assessment of Psoriasis and Psoriatic Arthritis (GRAPPA) group (Ritchlin et al, 2009). In addition, for the genotyping of *NOX4* variants and for ROS measurement in blood samples we recruited psoriasis ($n = 1382$) and psoriatic arthritis patients ($n = 492$) and normal healthy controls ($n = 484$). Caucasian origin was ascertained through ethnicity SNP genotyping (Giardina et al, 2008). Blood samples from one PAM patient (PAM12, male, 60 years old), one PsA patient carrying one of the rare variants found in PAM (PsA961, male, 39 years old), one PsA patient not carrying any of the variants investigated (PsA77, male, 43 years old) and age-gender-matched healthy controls were used for in vitro osteoclast differentiation.

### DNA isolation

DNA from whole blood was purified by Gentra Puregene Blood Kit (158489, Qiagen, USA). Briefly, three volumes RBC Lysis Solution was added to blood and centrifuged at 4000×*g* for 10 min to pellet the white blood cells (WBS), supernatant was discarded. The WBS were lysed with one volume of cell lysis solution by vortexing. The cell lysates were treated with RNase A, and proteins were precipitated. The supernatant containing DNA was then extracted

with isopropanol, followed by ethanol precipitation. After purification, the DNA was measured by Qubit.

### Whole-genome and whole-exome sequence (WGS and WES)

We applied whole-genome sequencing (WGS) and whole-exome sequencing (WES) to 5 and 56 PAM patients, respectively. In addition, the parents of one PAM patient were sequenced by WGS. We applied Somalier, a tool to measure relatedness in cohorts (https://github.com/brentp/somalier) to identify cryptic relatedness among all the samples (Fig. EV3).

WGS was performed at the Science for Life Laboratory's (SciLifeLab) national genomics infrastructure (NGI). The sequencing libraries were constructed using 1 μg of high-quality genomic DNA using the Illumina (San Diego, CA) TruSeq PCR-free kits (350 bp insert size) and sequenced on a single Illumina HiSeqX PE 2x150bp lane. WES was conducted at Uppsala's SNP & SEQ technological platform. We utilized 300 ng of genomic DNA for WES; the DNA quality was determined using the FragmentAnalyzer, and the DNA concentration was determined using the Qubit/Quant-iT test. The sequencing libraries were constructed using the Twist Human Core Exome (Twist Bioscience), and the sequencing was carried out in a single S4 lane using the Illumina NovaSeq equipment and v1 sequencing chemicals (150 cycles paired-end). The data were processed, and the sequence reads were aligned to the human genome build GRCh37 Single nucleotide variants (SNVs) and insertions/deletions (INDELs) were called using the GATK v3.8. and v 4.1.4.1 pipeline and the called variants were annotated using VEP (v.91). The variants were loaded into the GEMINI database to query and filter the variants. Variants with a minor allele frequency (MAF) of <0.0001 were filtered for further investigation. The variants found were inspected manually with the integrative genomics viewer (IGV) tool in the other patients. The impact of variants was evaluated using the prediction tools SIFT, Polyphen-2, CADD and GERP + +. Selected variants were examined manually in the BAM files using Integrated Genomics Viewer.

### Structural variants

Structural variants (SV) were analyzed using FindSV, a pipeline that performs SV detection using TIDDIT and CNVnator, as well as variant filtering and annotation using VEP and SVDB (Eisfeldt et al, 2017). Selected variants were visualized by the Integrative Genomics Viewer (IGV) tool.

### SNP genotyping

Genotyping of three Single Nucleotide Polymorphisms (SNPs) within the *NOX4* gene (rs781430033, rs144215891, and rs765662279) was performed by using allele-specific Taqman MGB probes labeled with fluorescent dyes FAM and VIC (Applied Biosystems, Foster City, CA, USA), according to the manufacturer's protocols. Allelic discrimination was made with the QuantStudioTM Real-Time PCR Software (Applied Biosystems). All three mutations had to be custom-made by using Custom TaqMan®Assay Design Tool (https://www.thermofisher.com/order/custom-genomic-products/tools/cadt/); The success rate for genotyping

exceeded 99% for all SNPs in the total sample set. We ran ten percent of the samples as duplicates to identify errors in genotyping and we could confirm assay accuracy of all three variations by WES and WGS of 61 PAM samples. The PCR procedure has been done using a total volume of 10 μl containing 15 ng of genomic DNA, 5 ul TaqMan® Universal PCR Master Mix (2×) and 0.5 μl TaqMan® genotyping assay mix (20×). Sequences of TapMan probes and primers are listed in Appendix Table S3. Following an initial denaturation step at 50 °C for 2 min and 95 °C for 10 min starting all PCR procedures comprised 40 cycles of denaturation at 95 °C for 15 s, and primer annealing at 60 °C (55 °C for rs10065172) for 1 min and saved at 4 °C. We performed an endpoint plate read comprising the last step with an increasing temperature to a maximum of 60 °C (1.6 °C per second) and accompanying measurement of fluorescence intensity on a real-time PCR on the QuantStudio 7 Flex Real-Time PCR System Instrument.

## Sanger sequencing

Genomic DNA from peripheral blood samples was extracted by standard procedures. Sanger sequencing was performed by KIGene using the ABI 3730 PRISM® DNA Analyzer (Zianni et al, 2006). The primers used are shown in Appendix Table S4.

## Expression constructs and stable HEK293 cell lines

To test the effect of the variants in cells, we obtained plasmid -pcDNA3.1-hNox4 (#69352) from the Addgene repository. Primers with the alternate alleles for each SNP were designed using the "QuikChange Primer Design" (Agilent technologies) platform. Then, $NOX4^{Y512IfsX20}$, $NOX4^{V369F}$ and $NOX4^{Y512C}$ variants were introduced to the construct by using QuikChange XL Site-Directed Mutagenesis Kit (Agilent) according to manufacturer instructions with the primer pairs in Appendix Table S5, which were transformed into Escherichia coli and identified by Sanger dideoxy sequencing (Appendix Table S6). To obtain stable transfectants, we linearized plasmids with 1 μl BglII Enzyme (10 unit) and 5 μl 10× NEB buffer with the incubation at 37 °C for 15 min and 65 °C for 20 min. Human embryonic kidney 293 (HEK293) cells were kindly provided by Stefano Gastaldello (Karolinska Institutet, Stockholm, Sweden). HEK293 cells were cultured in Dulbecco's modified Eagle medium (DMEM; Gibco) supplemented with 10% fetal bovine serum (FBS; Gibco) and 100 μg/ml Primocin (InvivoGen) at 37 °C in a humidified 5% $CO_2$ atmosphere, and periodically checked for mycoplasma contamination. HEK293 cells were transfected with pcDNA3.1-hNOX4 and three constructs carrier NOX4 variants by the Lipofectamine 2000 Reagent (Thermo Fisher Scientific, USA), and G418 at 200 μg/ml was used as positive cell selection. Culture media containing the selection antibiotic was changed every 2–3 days until Geneticin®-resistant foci were identified. Next, we screened single-colony cells in the 96-well tissue culture plate and expanded the selected cells for future use.

## Quantitative real-time PCR analysis

The extraction of total RNA from HEK293 stable cell lines was isolated by RNeasy mini kit (QIAGEN), and cDNA was reversed with Maxima First Strand cDNA Synthesis Kit (Thermo Fisher Scientific). Real-time quantitative PCR (RT-qPCR) was performed with SYBR® Green Master Mix according to the manufacturer's protocol. Primer sequences are provided at Appendix Table S2. The $2 - \Delta\Delta Ct$ method was utilized to achieve comparative quantification of the gene of interest between the two genotypes using actin as a reference gene.

## Western blotting

HEK293 cells were harvested and lysed in RIPA lysis with 1X Halt™ Protease and Phosphatase Inhibitor Cocktail (Thermo Fisher Scientific). The concentration of total protein was determined using the BCA Protein Assay Kit (Thermo Fisher Scientific). In total, 10 μg or 20 μg of proteins were loaded in 10% SDS-PAGE gel and transferred to PVDF membranes. Membranes were blocked in Tris-buffered saline containing 5% skim milk for 1.5 h at room temperature. Then incubated at 4 °C overnight with recombinant anti-NADPH oxidase 4 antibody (1:2000, ab133303, abcam), and mouse anti-GAPDH monoclonal antibody was used for normalization (1:3000, 60004-1, Proteintech Group Inc). Immunoblots of protein bands were visualized with ECL (1705060, Clarity™ Western ECL Substrate, Biorad), and proteins were quantified with ImageJ software. The data are presented as mean ± SD of independent experiments performed in triplicate.

## Osteoclast studies from patient-derived peripheral blood mononuclear cells (PBMCs)

To study the effect of the mutations on osteoclasts differentiation in cell culture, we obtained patient-derived mononuclear cells. PBMCs were isolated from whole blood using Ficoll-Paque density centrifugation. For positive selection of the osteoclast precursors, i.e., the CD14+ mononuclear cells, the EasySepTM Human CD14 positive Selection kit II was used according to the manufacturer's instructions. Purified CD14+ cells were seeded in 24-well and 96-well plates containing Gibco DMEM supplemented with 10% FBS, 0.2% PrimocinTM and macrophage colony-stimulating factor (M-CSF) (20 ng/mL; R&D systems; USA) and receptor activator of nuclear factor kappa-B ligand (RANKL) (2 ng/mL; R&D systems; USA) to induce osteoclastogenesis. Every third day, media was refreshed. The osteoclasts were fixed and stained with tartrate-resistant acid phosphatase (TRAP)-positive cells based on a leukocyte acid phosphatase kit (387A; Sigma; USA) according to the manufacturer's instructions. TRAP-stained cells containing three or more nuclei were defined as osteoclasts (Makitie et al, 2021).

## Measurement of ROS by electron paramagnetic resonance (EPR)

The levels of ROS in the human blood and cultured cells were measured by EPR Spectroscopy (Zheng et al, 2022). Following ~36 h after transfection, cell culture media was removed, and the cells were rinsed twice with PBS. Seven hundred microliters of cyclichydroxylamine (CMH, 200 μM) in EPR-grade Krebs HEPES buffer supplemented with 25 μM Deferoxamine (DFX) and 5 μM diethyldithiocarbamate (DETC) were added to the cells and were incubated for 30 min at 37 °C. The cells are collected in the 1-mL syringes and frozen in liquid nitrogen prior to measurement. To

analyze ROS levels in human blood, blood samples were incubated with CMH spin probe as mentioned above and ROS was measured using the EPR spectrometer (Noxygen, Elzach, Germany). ROS levels were converted to the concentration of CP radical using the standard curve method. Briefly, blood samples were combined with cyclichydroxylamine (CMH) spin probe and ROS was measured by a CP radical standard curve, using EPR spectrometer (Noxygen, Elzach, Germany).

## Measurement of ROS production

2′,7′-dichlorofluorescein diacetate (4091-99-0, DCFH-DA, Sigma) was used as a sensitive and rapid identification of ROS in response to oxidative metabolism. First, we reconstituted in DMSO for stock, and then DCFH-DA S0033) was diluted with the serum-free cell culture medium. After washing osteoclasts with PBS twice at designated time points—day 8 and day 12, osteoclasts (at the density of $1 \times 10^5$ cells/well) in a 96-well plate were incubated with $10 \mu M$ DCFH-DA in the incubator for 30 min and thereafter immediately analyzed using a fluorescence microscope (magnification ×10; EVOS™ FL, Invitrogen). The relative fluorescence intensity of DCFH-DA was analyzed using ImageJ.

## Measurement of $H_2O_2$

Intracellular levels of $H_2O_2$ were detected utilizing the BioTracker Green $H_2O_2$ live-cell dye according to the manufacturer's recommendation (SCT039, Sigma). For HEK293 stable cell lines, as well as osteoclasts derived from patients and controls, the culture medium was removed, and cells were rinsed twice with Hank's Balanced Salt Solution (HBSS). Subsequently, a 1 mM concentration of the solution was added and the cells were then incubated for 20 min at 37 °C, 5% $CO_2$. After removing the solution, cells underwent two additional rinses with HBSS. The visualization of living cells was conducted utilizing a fluorescence microscope (EVOS™ FL, Invitrogen).

## Zebrafish assay for oxidative stress

The pcDNA3.1-h*NOX4*, *NOX4*[Y512IfsX20], *NOX4*[V369F] and *NOX4*[Y512C] plasmids were linearized by restriction digestion with XhoI enzyme, and capped mRNA was transcribed in vitro using the mMESSAGE mMACHINE kit (Ambion, Thermo Fisher Scientific, Waltham, MA, USA). Zebrafish embryos (AB strain) at 1–2-cell stage were co-injected with *NOX4* mRNA and an antisense oligonucleotide used to knockdown endogenous *nox4* expression (*nox4* atg MO) (300 pg). For in vivo ROS detection, 30 h post fertilization, old embryos were exposed to 20 μM 2′,7′-dichlorofluorescein diacetate (DCFH-DA, Sigma-Aldrich) for 1 h at 28.5 °C in the dark followed by washing with embryo water a minimum of three times (Zhang et al, 2014). ROS production was visualized by mounting each embryo in a drop of low melting agarose (Hirsinger and Steventon, 2017) and imaged using a confocal microscope (Zeiss LSM700 coupled with a water-dripping lens). Importantly, every embryo was imaged using the same settings (e.g., laser intensity and optical slice thickness). The average fluorescence density (normalized to area) was analyzed using ImageJ. To this end, maximum intensity projections were produced and the total fluorescent intensity within the defined area was quantified. For each experimental group 10–16 embryos were quantified, and the experiment was repeated three times.

## Statistics

Blinding was only applied for counting the number of TRAP-positive osteoclasts by two independent individuals. The assignment of TRAP figures was randomized to eliminate systematic errors, ensuring an unbiased analysis where each figure is represented in the evaluation. Images of cells located in the center of the wells were included in the analysis, while images of cells at the borders, where the distribution was uneven, were excluded.

Quantification and Statistical Analysis were performed using GraphPad Prism9. All experiments were performed with at least three independent biological replicates and expressed as the means ± standard deviation (SD). Student's $t$ test was applied to assess the statistical differences between experimental groups. The one-way analysis of variance (ANOVA) was used to assess the statistically significant differences between the means of three unrelated groups. Multiple comparisons were evaluated for all pairs of means by Two-way ANOVA with Tukey's correction. $P < 0.05$ was considered significant (*$P < 0.05$, **$P < 0.01$, ***$P < 0.001$, ****$P < 0.0001$).

---

**The paper explained**

**Problem**

Psoriatic arthritis mutilans (PAM) is the most severe and rare form of psoriatic arthritis. PAM is characterized by severe destruction of the joints leading to osteolysis in hands and feet. There is no cure for PAM. Despite the severity, the etiology of PAM remains poorly understood. Our study aimed to unravel genetic factors and underlying biological mechanisms contributing to PAM.

**Results**

The study was performed on the PAM Nordic cohort consisting of 61 well-characterized PAM patients. We report three rare genetic variants in four patients in the *NOX4* gene (*NOX4*[Y512IfsX2], *NOX4*[V369F], and *NOX4*[Y512]). All variants are potentially damaging and located in important domains for the function of NOX4. NOX4 is an enzyme that produces reactive oxygen species (ROS), mainly hydrogen peroxide ($H_2O_2$). Two of the variants found affect the same codon in the NADPH-binding domain, and the third variant affects the FAD binding domain. Both domains are important for the proper function of the enzyme. Functional analysis revealed that the rare variants led to increased ROS in cell models, zebrafish embryos, and patient-derived osteoclasts. Furthermore, elevated ROS levels were confirmed in the blood of specific PAM patients, indicating the potential role of ROS in PAM pathogenesis.

**Impact**

The identification of *NOX4* variants in PAM provides novel insights into the genetic basis of this severe form of psoriatic arthritis. The study expands our understanding of the role of NOX4 and ROS pathways (specifically $H_2O_2$) in joint destruction, opening avenues for targeted therapeutic interventions. In addition, the findings may have broader implications for understanding and treating other conditions associated with NOX4 and ROS dysregulation. The study contributes valuable information towards precision medicine approaches for PAM, opening the possibility for a potential therapeutic target.

## Study approval

The study was approved by ethical review boards at each institution and conducted according to the Declaration of Helsinki Principles. Written informed consent was obtained from all the participants in the study. Ethical permits:2007/1088-31/4. D.nr:00-448, 2008/4:5, Dnr 02-241, and Dnr 2022-04253-02. The Stockholm Ethical Board for Animal Experiments authorized standard operating procedures for all treatments involving zebrafish (Ethical approval: Dnr 14049-2019).

## Graphics

Synopsis image was created with BioRender.com.

For more information please see: (1) gnomAD, https://gnomad. broadinstitute.org/; (2) PolyPhen-2, http://genetics.bwh.harvard.edu/ pph2/; (3) SIFT, https://sift.bii.a-star.edu.sg; (4) CADD, https:// cadd.gs.washington.edu/info; (5) SweGen, https://swefreq.nbis.se/ dataset/SweGen; (6) dbSNP, https://www.ncbi.nlm.nih.gov/snp/.

# Data availability

This study includes no data deposited in external repositories, as the patient whole-genome and whole-exome sequencing data cannot be freely available due to consent agreement restrictions.

# Peer review information

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

## Acknowledgements

We gratefully acknowledge the patients and controls for participation in this project. We also would like to thank Helena Griehsel for helping in taking samples from patients and Jose Laffita-Mesa, Anton Tornqvist and Xiaoyuan Ren for technical assistance. Our gratitude extends to Xinyue Zhao for meticulous support in photo editing and to Andrea Bieder for providing critical editorial feedback on the manuscript. The authors acknowledge support from the National Genomics Infrastructure in Stockholm funded by Science for Life

Laboratory, the Knut and Alice Wallenberg Foundation and the Swedish Research Council, and SNIC/Uppsala Multidisciplinary Center for Advanced Computational Science for assistance with massively parallel sequencing and access to the UPPMAX computational infrastructure. We also acknowledge the support provided by the Biomedicum Imaging Core and Zebrafish Core Facility employees in maintaining the microscopes and caring for the zebrafish. This work was supported by Hudfonden (grants 3378, 3227, and 2808 to ITP, MS and to PN), Swedish Rheumatism Association, Reumatikerförbundet (R-968063 to ITP), Konung Gustaf V:s 80-årsfond (FAI-2021-0819a to ITP), Psoriasisfonden to MS and ITP, The European academy of dermatology and venereology (EADV) (PPRC-2022-40 to ITP), Doctoral scholarship KI-China Scholarship Council (CSC) programme to SW, Stiftelsen Sällsyntafonden to SW, FT, and RV.

## Author contributions

Sailan Wang: Conceptualization; Formal analysis; Funding acquisition; Validation; Investigation; Visualization; Methodology; Writing—original draft; Project administration; Writing—review and editing. Pernilla Nikamo: Formal analysis; Funding acquisition; Validation; Methodology; Project administration; Writing—review and editing. Leena Laasonen: Resources; Writing—review and editing. Bjorn Gudbjornsson: Resources; Writing—review and editing. Leif Ejstrup: Resources; Writing—review and editing. Lars Iversen: Resources; Writing—review and editing. Ulla Lindqvist: Resources; Writing—review and editing. Jessica J Alm: Supervision; Methodology; Writing—review and editing. Jesper Eisfeldt: Data curation; Software; Formal analysis; Writing—review and editing. Xiaowei Zheng: Methodology; Writing—review and editing. Sergiu-Bogdan Catrina: Methodology; Writing—review and editing. Fulya Taylan: Data curation; Software; Formal analysis; Methodology; Writing—review and editing. Raquel Vaz: Formal analysis; Supervision; Visualization; Methodology; Writing—original draft; Writing—review and editing. Mona Ståhle: Conceptualization; Resources; Supervision; Funding acquisition; Investigation; Methodology; Writing—original draft; Project administration; Writing—review and editing. Isabel Tapia-Paez: Conceptualization; Formal analysis; Supervision; Funding acquisition; Validation; Investigation; Visualization; Methodology; Writing—original draft; Project administration; Writing—review and editing.

## Funding

## Disclosure and competing interests statement

LI has served as a consultant and/or paid speaker for and/or participated in clinical trials sponsored by: AbbVie, Almirall, Amgen, Astra Zeneca, BMS, Boehringer Ingelheim, Celgene, Centocor, Eli Lilly, Janssen Cilag, Kyowa, Leo Pharma, Micreos Human Health, MSD, Novartis, Pfizer, Regranion, Samsung, Union Therapeutics, UCB. The remaining authors declare no competing interests.

# Expanded View Figures

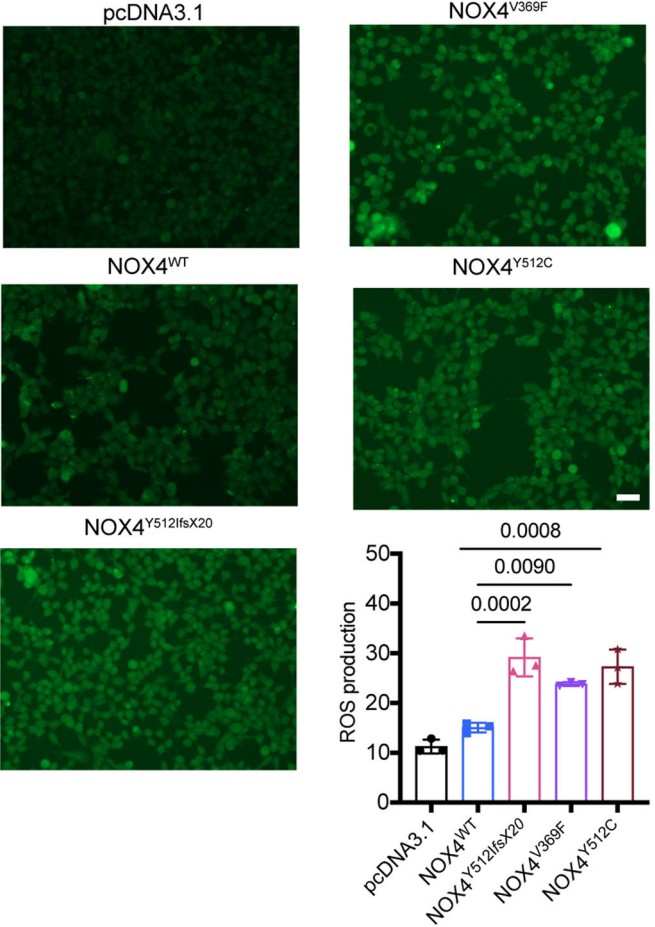

**Figure EV1.   Elevated ROS levels in HEK293 stable transfected cell lines expressing *NOX4* variants.**

HEK293 cells were subjected to 12 h of serum starvation after stably transfection with following plasmids: pcDNA3.1, *NOX4^wt^*, *NOX4^Y512IfsX20^*, *NOX4^Y512C^*, and *NOX4^V369F^*. Fluorescence imaging was conducted following the incubation with 10 µM DCFH-DA. Representative photomicrographs of the fluorescence are displayed and quantification of the mean fluorescence intensity was performed with ImageJ software. $N = 3$. Scale bars: 100 µm. Data information: DCFH-DA 2′,7′-dichlorofluorescein diacetate. Data presented as mean ± SD. $N =$ biological replicates. The *P* value was calculated by the ordinary one-way ANOVA multiple comparisons with Turkey correction of multiple hypothesis tests. Source data are available online for this figure.

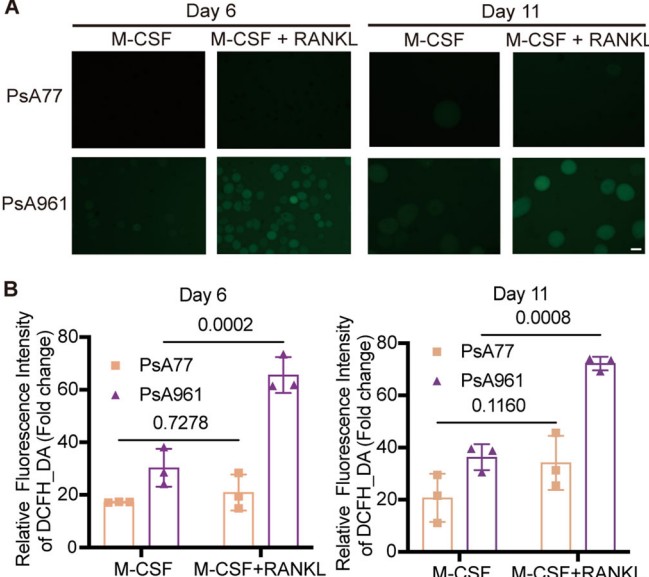

**Figure EV2. Impact of ROS on osteoclasts differentiation from PsA961 (NOX4^{Y512C}).**

(A) Cells were cultured in the presence of M-CSF or M-CSF and RANKL. Visualization of ROS detected by DCFH-DA on Day 6 and Day 11, indicating a higher ROS effect in PsA961-differentiated osteoclasts. Scale bar: 100 μm. (B) The relative fluorescence intensity of ROS was quantified by GraphPad Prism9.0.0. N = 3. Data information: DCFH-DA 2′,7′-dichlorofluorescein diacetate. N = biological replicates. For graph (B), error bars in figure represent mean ± SD (two-way ANOVA with Tukey's multiple comparisons tests). Source data are available online for this figure.

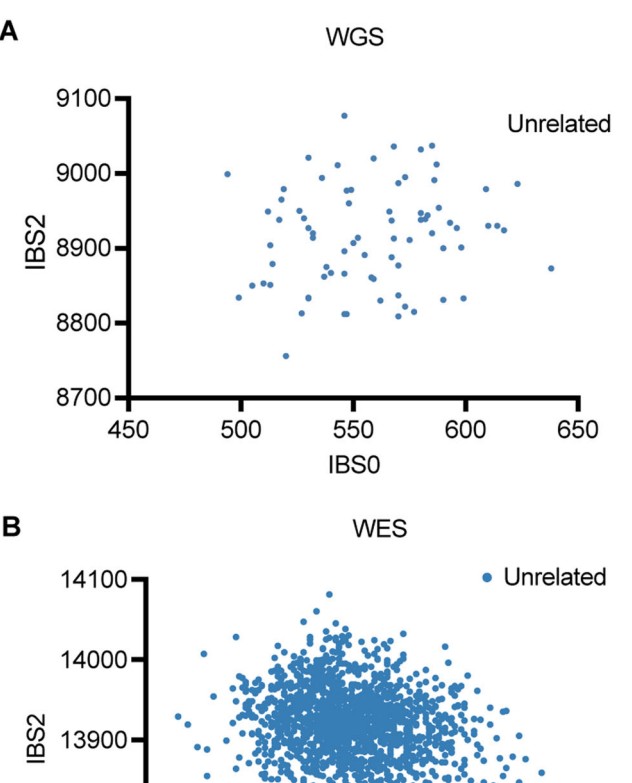

**Figure EV3.   Relatedness plot for WGS (A) and WES (B) samples.**

(A) Relatedness plot for WGS samples. Six samples from the SweGene database (https://swefreq.nbis.se/) were added to the WGS analysis. Each dot represents a pair of samples. (B) Relatedness analysis of all WES samples. Data information: IBS0 is the number of sites where 1 sample is homozygous for the reference allele and the other is homozygous for the alternate allele. IBS2, is the count of sites where a pair of samples were both homozygous or both heterozygous. WGS Whole-Genome Sequencing, WES Whole-Exome Sequencing. Source data are available online for this figure.

