## [Peer Review File · EMBO Molecular Medicine]

Rare coding variants in NOX4 link high ROS levels to psoriatic arthritis mutilans

Sailan Wang, Pernilla Nikamo, Leena Laasonen, Bjorn Gudbjornsson, Leif Ejstrup, Lars Iversen, Ulla Lindqvist, Jessica Alm, Jesper Eisfeldt, Xiaowei Zheng, Sergiu-Bogdan Catrina, Fulya Taylan, Raquel Vaz, Mona Ståhle, and Isabel Tapia-Paez

Corresponding author: Isabel Tapia-Paez (isabel.tapia@ki.se)

Review Timeline:

Submission Date:	14th Jun 23
Editorial Decision:	14th Jul 23
Revision Received:	8th Dec 23
Editorial Decision:	11th Jan 24
Revision Received:	19th Jan 24
Accepted:	26th Jan 24

Editor: Lise Roth

Transaction Report:

14th Jul 2023

Dear Dr. Tapia-Paez,

Thank you for the submission of your manuscript to EMBO Molecular Medicine. We have now received feedback from the three reviewers who agreed to evaluate your manuscript. As you will see below, the reviewers raise substantial concerns on your work, which unfortunately preclude its publication in EMBO Molecular Medicine in its current form.

The reviewers find that the question addressed by the study is of potential interest, however they remain unconvinced that some of the major conclusions are sufficiently supported by the data. In particular, a critical point raised by referee #3 relates to the claim that Nox4 generates superoxide while it has been shown to instead release hydrogen peroxide.

If you feel you can satisfactorily address this point and the other issues raised by the referees, you may wish to submit a revised version of your manuscript. Please attach a covering letter giving details of the way in which you have handled each of the points raised by the referees. A revised manuscript will once again be subject to review, and we cannot guarantee at this stage that the eventual outcome will be favorable. EMBO Molecular Medicine encourages a single round of revision only and therefore, acceptance or rejection of the manuscript will depend on the completeness of your responses included in the next, final version of the manuscript. For this reason, and to save you from any frustrations in the end, I would strongly advise against returning an incomplete revision.

As revising the manuscript according to the referees' recommendations appears to require a lot of additional work and experimentation, and given the potential interest of your findings, we are ready to extend the deadline to 6 months.

We require:

- 1) A .docx formatted version of the manuscript text (including legends for main figures, EV figures and tables). Please make sure that the changes are highlighted to be clearly visible.
- 2) Individual production quality figure files as .eps, .tif, .jpg (one file per figure). For guidance, download the 'Figure Guide PDF' (<https://www.embopress.org/page/journal/17574684/authorguide#figureformat>).
- 3) At EMBO Press we ask authors to provide source data for the main figures. Our source data coordinator will contact you to discuss which figure panels we would need source data for and will also provide you with helpful tips on how to upload and organize the files.
- 4) A .docx formatted letter INCLUDING the reviewers' reports and your detailed point-by-point responses to their comments. As part of the EMBO Press transparent editorial process, the point-by-point response is part of the Review Process File (RPF), which will be published alongside your paper.
- 5) A complete author checklist, which you can download from our author guidelines (<https://www.embopress.org/page/journal/17574684/authorguide#submissionofrevisions>). Please insert information in the checklist that is also reflected in the manuscript. The completed author checklist will also be part of the RPF.
- 6) Please note that all corresponding authors are required to supply an ORCID ID for their name upon submission of a revised manuscript.
- 7) It is mandatory to include a 'Data Availability' section after the Materials and Methods. Before submitting your revision, primary datasets produced in this study need to be deposited in an appropriate public database, and the accession numbers and database listed under 'Data Availability'. Please remember to provide a reviewer password if the datasets are not yet public (see <https://www.embopress.org/page/journal/17574684/authorguide#dataavailability>). In case you have no data that requires deposition in a public database, please state so in this section. Note that the Data Availability Section is restricted to new primary data that are part of this study.
- 8) For data quantification: please specify the name of the statistical test used to generate error bars and P values, the number (n) of independent experiments (specify technical or biological replicates) underlying each data point and the test used to

calculate p-values in each figure legend. The figure legends should contain a basic description of n, P and the test applied. Graphs must include a description of the bars and the error bars (s.d., s.e.m.). Please provide exact p values.

13) Author contributions: CRediT has replaced the traditional author contributions section because it offers a systematic machine readable author contributions format that allows for more effective research assessment. Please remove the Authors Contributions from the manuscript and use the free text boxes beneath each contributing author's name in our system to add specific details on the author's contribution. More information is available in our guide to authors.

16) As part of the EMBO Publications transparent editorial process initiative (see our Editorial at <http://embomolmed.embopress.org/content/2/9/329>), EMBO Molecular Medicine will publish online a Review Process File (RPF) to accompany accepted manuscripts.

In the event of acceptance, this file will be published in conjunction with your paper and will include the anonymous referee reports, your point-by-point response and all pertinent correspondence relating to the manuscript. Let us know whether you agree with the publication of the RPF and as here, if you want to remove or not any figures from it prior to publication. Please note that the Authors checklist will be published at the end of the RPF.

EMBO Molecular Medicine has a "scooping protection" policy, whereby similar findings that are published by others during review or revision are not a criterion for rejection. Should you decide to submit a revised version, I do ask that you get in touch

after three months if you have not completed it, to update us on the status.

I look forward to receiving your revised manuscript.

Yours sincerely,

Lise Roth

***** Reviewer's comments *****

Referee #1 (Remarks for Author):

Psoriatic arthritis mutilans (PAM) is the most severe form of PsA, with unknown underlying mechanisms including genetic backgrounds. The authors found rare variants in NOX4 in 4/61 PAM patients. NOX4 is a transmembrane enzyme expressed in many cell types including keratinocytes and osteoclasts, functionally involved in the production of ROS. Additional variants were also found in other genes related with ROS production. In fact, patients with NOX4 rare variants showed up-regulation of ROS in their peripheral blood, CD14+ cells from one of patient showed increased differentiation into osteoclasts. Upon hypothesizing that those rare variants of NOX4 might increase the levels of ROS, the authors conducted several in vitro and in vivo models. This study is clinically intriguing in that at least a subset of patients suffering from the most severe form of PsA had rare variant in the pathway related with ROS production. The manuscript is well-written and intensively evaluated the functional aspects of the variant genes. This reviewer rises several concerns.

From in vitro experiments, variant forms of NOX4 seem to be GOF ones, but in-silico prediction suggested they are damaging. It is hard to understand why frame-shift mutation in the functional cytoplasmic domain increases the function of the gene. Is it possible that other NOX family proteins are over-expressed in those with NOX4 variants?
Is any SNP or rare variants with known eQTL effect on NOX4 detected in their patients (other than intronic SNP rs11018628)?
Did the author find any SNP or rare variant in the promotor/enhancer lesion of NOX4 gene in investigated patients?

Figure 3

In overexpression system, transcript levels largely depend on the structures of each expression vectors. It is unclear how those artificial expression vectors represent phenomena found in their patients. Was there statistically significant difference in ROS levels between pcDNA3.1 and NOX4WT?

Figure 4

How the authors think of the effect of TNF-inhibitors on ROS production? When etanercept was predicted to increase ROS production, why adalimumab was considered to decrease that?

Figure 5

Osteoclast differentiation from peripheral blood is interesting, but evaluated in only one patient.

Referee #2 (Comments on Novelty/Model System for Author):

Potentially, the findings improve the understanding of this severe form of psoriatic arthritis.

Referee #2 (Remarks for Author):

In this manuscript, Wang S. et al. studied a cohort of Nordic patients (Sweden, Denmark and Norway) with psoriatic arthritis mutilans (PAM). By using of next generation sequencing (whole exome sequencing and whole genome sequencing), they found

rare variants in NOX4 gene, which are characterized in overexpression system and animal model (zebrafish) in terms of reactive oxygen species (ROS) production. The authors concluded that these variants cause a high ROS production. In addition, they found other variants in some genes related with ROS/NOX4 pathway. See my comments:

-The model to find rare variants are missing in this article: autosomal recessive, autosomal dominant or X-recessive pattern? MAF?

-Familial segregation is also missing here. I assumed regarding the electropherogram that variants of patients are heterozygous.

-Expression of RNA was evaluated in HEK293 cell lines. However, the authors do not show protein expression.

-The authors should specify that are the cells used for testing ROS production: whole blood, peripheral mononuclear cells, peripheral monocytes, neutrophils, osteoclast differentiation?

-Samples of PAM12 were used for the differentiation of osteoclasts and the evaluation of ROS production. What is here the connexion? She/he is mutated in NOX4?

Referee #3 (Remarks for Author):

Wang et al have applied whole-genome sequencing (n=5) and whole exome sequencing (n=56) to search for potentially disease relevant gene polymorphisms in patients with Psoriatic arthritis mutilans (PAM); an infrequent and severe variant of psoriatic arthritis. They found rare variants in the NADPH oxidase 4 (NOX4) in four patients.

NOX4, a member of the NOX-family of enzymes that functions as the catalytic subunit of the NADPH oxidase complex, is known to be involved in the production of reactive oxygen species (ROS). The authors have tested in in vitro and in vivo models whether the identified variants are involved in enhancing ROS release. Specifically, they are presenting data from osteoclasts of 1 patient with PAM, from stably transfected cell lines, from direct measurements of superoxide in patient blood samples and from zebra fish models. They conclude that NOX4 is a susceptibility gene in PAM and contributes to disease through production of high levels of superoxide, specifically in osteoclast.

Comments:

1. PAM is a disabling subtype of psoriatic arthritis and genetic risk factors have not yet been identified. Therefore, the current study is timely and potentially impactful.

2. The authors have available a cohort of n=61 PAM patients that have been clinically well phenotyped. This is a valuable resource.

3. While it is straightforward to speculate that a polymorphism in a NOX gene is involved in ROS production, the mechanistic data presented in the current manuscript remain unconvincing.

4. As a general rule, the study cohorts for the mechanistic experiments are very small, often consisting of only one patient.

5. Nox4 is unusual as it releases hydrogen peroxide (H₂O₂) in contrast to Nox1-Nox3 and Nox5, which release superoxide (Nisimoto et al, Biochemistry 2014). Curiously, the authors present several data sets showing that NOX4 generates superoxide and the title of the manuscript implicates NOX4 in inducing high superoxide levels. This is a critical claim of the manuscript and requires clarification.

6. Not only does Nox4 emit a different pattern of ROS, but there are fundamental differences between NOX4 and the other members of the NOX family. Specifically, its subcellular localizations, tissue distribution and influence over signaling pathways is different from the other Nox enzymes. Nox4 is recognized for its role in oxygen sensing, vasomotor control, cellular proliferation, differentiation, migration, apoptosis, senescence, fibrosis, and angiogenesis.

The current manuscript is focused exclusively on the expression of NOX4 in osteoclastic cells, a subtype of myeloid cells. This appears to be a missed opportunity and it is not clear how the authors arrived at this focus. Here, an unbiased approach would make more sense.

7. The Human Protein Atlas provides data about the expression of NOX4 in different cell types.

By far the highest expression of NOX4 occurs in proximal tubular cells in the kidney. To a much lower degree is NOX4 found in fibroblasts and smooth muscle cells in different organs. It is explicitly noted that immune cells, including the myeloid cell types, are low for NOX4 expression. Here, it may be important to consider how NOX4+ proximal tubular cells in the kidney could have a potential role in PAM.

8. NOX4 is overexpressed in some tumors and has been implicated in regulating oncogenic metabolic adaptations. This is another important aspect of NOX4 biology that has not been considered in the current manuscript.

9. Figure 3: The authors are showing highly elevated NOX4 RNA expression in HEK293 cell lines transfected with a vector

carrying wt or variant NOX4. Did they control for how many copies of the vector each of the cell lines contains?

10. Figure 3: The expression of NOX4 RNA appears to be massively higher if HEK293 cells are transfected with vectors containing NOX4(Y512IfsX20), NOX4(Y512C), and NOX4(V369F). If correct, this is an interesting observation. Do the authors know what the underlying mechanism is?

The authors state that the polymorphisms affect the NADPH binding domain of the enzyme (Line 309). Why would that lead to a change in RNA abundance? This is difficult to understand.

11. Figure 3B. The authors show higher superoxide expression in the cell lines transfected with NOX4 variants. NOX4 releases hydrogen peroxide (H₂O₂) in contrast to Nox1-Nox3 and Nox5, which release superoxide (Nisimoto et al, Biochemistry). Here, it would be critical to show production of hydrogen peroxide.

12. Line 426-428: "The results suggest that the variant NOX4 (Y512C) affecting the NADPH binding domain is responsible for the elevated superoxide production."

Are the authors proposing that the rare variants of NOX4 change emission of the ROS pattern? To support that claim, they would need to show precise measurements of H₂O₂ and of superoxide (ideally in a membrane-free system).

13. Line 429-430: "It should be noted here that we did not find any NOX4 mutations in PAM12 and PAM37, nor mutations in any other genes related to NOX4."

Nevertheless, the authors chose PAM12 and PAM37 to examine peripheral blood ROS measurements. PAM12 appears to produce high levels and PAM37 appears to produce levels indistinguishable from controls.

- The authors should clarify what "peripheral blood" is. Is this serum? Plasma? Does it contain cells?

- It is unclear why they examined patients that do not carry the variant.

- To examine a single patient is insufficient to draw any conclusions. Here, they need to examine patient cohorts of sufficient size and include patients with other inflammatory diseases as controls. High ROS production has been implicated in a variety of inflammatory diseases. To support their claim, the authors need to link the high ROS production to NOX4.

14. In some experiments, the authors have used DCFH-DA for "ROS quantification". Quantifying specific subtypes of ROS is challenging, particularly for H₂O₂ (see recently developed Guidelines; Murphy et al, Nature Metabolism, 2022). DCFH does not directly react with H₂O₂ to form the fluorescent product, DCF. Therefore, DCF fluorescence cannot be used as a direct measure of H₂O₂. Experiments utilizing DCFH-DA should be interpreted with caution.

Referee #1 (Remarks for Author):

Psoriatic arthritis mutilans (PAM) is the most severe form of PsA, with unknown underlying mechanisms including genetic backgrounds. The authors found rare variants in NOX4 in 4/61 PAM patients. NOX4 is a transmembrane enzyme expressed in many cell types including keratinocytes and osteoclasts, functionally involved in the production of ROS. Additional variants were also found in other genes related with ROS production. In fact, patients with NOX4 rare variants showed up-regulation of ROS in their peripheral blood, CD14+ cells from one of patient showed increased differentiation into osteoclasts. Upon hypothesizing that those rare variants of NOX4 might increase the levels of ROS, the authors conducted several in vitro and in vivo models.

This study is clinically intriguing in that at least a subset of patients suffering from the most severe form of PsA had rare variant in the pathway related with ROS production. The manuscript is well-written and intensively evaluated the functional aspects of the variant genes. This reviewer rises several concerns.

Response: We thank the reviewer for the feedback and consider the comments to improve our manuscript. Please see below.

1) From in vitro experiments, variant forms of NOX4 seem to be GOF ones, but in-silico prediction suggested they are damaging. It is hard to understand why frame-shift mutation in the functional cytoplasmic domain increases the function of the gene. Is it possible that other NOX family proteins are over-expressed in those with NOX4 variants?

Response: Yes, indeed while the prediction typically associated frameshifts with LoF, is not always the case. There are instances where frameshift variants might lead to GoF as observed in conditions such as chondrodysplasia with *COL10A1* null and truncating mutations, leading to expansion of growth plates and chondrocyte differentiation (Ho *et al*, 2007) as well as in several tumors such as breast cancer (Mair *et al*, 2016) and Wilms tumor (Busch *et al*, 2014).

Interestingly, the frameshift variant (p.Y512lfsX20) that we found in our study exhibits a similar functional effect as the missense variant (p.Y512C) found in two PAM patients, which is located adjacent to it, affecting the same codon. This finding suggests that the location of these rare variants in the NADPH binding domain might be critical for the NOX4 enzymatic activity. The variants found could affect the protein folding, potentially prompting the cell to compensate by increasing gene expression. In the case of the frameshift variant, it shortens the protein by 46 amino acids, and the aberrant end of the protein might affect the NADPH binding, potentially increasing the generation of hydrogen peroxide.

Regarding the potential overexpression of other NOX family proteins in patients with *NOX4* rare variants, there is a possibility that due to compensatory mechanisms genes from the same family may try to compensate or may be triggered. In the experiments that we performed and presented in our manuscript, we measured the general ROS by using EPR and DCFH-DA. The results showed increase of ROS in all models, however the methods used could not identify specific ROS. Note that in the current manuscript we have added new in vitro assays with a probe detecting specifically hydrogen peroxide, the specific oxygen species released by NOX4 (Figures 3C-D and 6B-C).

In addition, identifying pathogenic variants among the approximately 4 million variants in our genome is a significant challenge. We performed WGS on five PAM Swedish patients followed by WES on the remaining 56 patients. There was no *a priori* hypothesis about specific genes involved in PAM. As most of our data is from WES, it's plausible that PAM patients may carry other variants (inter- or intra-genic) affecting NOX family genes (including NOX4) or other genes affecting ROS that might have eluded our analysis. Nevertheless, our experiments conducted in cell lines and in zebrafish models are designed to measure the effects of the specific variants found in *NOX4* compared to *NOX4* wild type. Therefore, we think that it is unlikely that the observed effects in the *in vitro* system are due to other *NOX* variants.

2) Is any SNP or rare variants with known eQTL effect on NOX4 detected in their patients (other than intronic SNP rs11018628)?

Response: We thank the reviewer for the comment. We checked the three variants found in PAM patients. However, they do not show any eQTLs in databases such as FIVEx, GTex and opentargets.

Regarding the eQTL SNPs for *NOX4*, the data available in databases is limited, covering adipose, artery, brain, fibroblasts, testis and skin tissues (**not bone**); when looking at the skin eQTLs for *NOX4*, all eQTLs are moderate negative (see the Figure R1 from GTEx. Skin eQTLs are marked by a green box).

Figure R1: eQTL for *NOX4* in GTEx.

The eQTL item color indicates the effect size attributed to the eQTL:

red	high positive
light red	moderate positive
light blue	moderate negative
blue	high negative

mixed	positive and negative effect in combined eQTL
-------	---

Furthermore, we also looked at whole blood eQTL data from <https://www.eqtlgen.org/phase1.html> and we observe that there are no cis-eQTLs nor trans-eQTLs linked to *NOX4* collected from 31,684 blood samples. Most eQTLs are linked to common variants in the population and therefore we may not make any assumptions that those would be potentially damaging for the rare disease PAM.

Did the author find any SNP or rare variant in the promotor/enhancer lesion of *NOX4* gene in investigated patients?

Response: Thank you for the question. We meticulously looked at the enhancer/promoter regions. But as our analysis encompassed WGS data for five patients and only coding WES data for most of the patients (n=56), we are very limited in this analysis. In the ENCODE database we looked at the regions with peaks in H3K4Me1, H3K4Me3 and H3K27Ac (see table R1). https://genome.ucsc.edu/cgi-bin/hgTracks?db=hg19&lastVirtModeType=default&lastVirtModeExtraState=&virtModeType=default&virtMode=0&nonVirtPosition=&position=chr11%3A89057522%2D89224653&hgsid=1706963600_zTGBwSMFhR1aChYnxNJEaXDUjXSW. In those regions we do not find any variants shared by at least two patients that would be very rare to investigate further; Instead, several indels with relatively high frequency in the population were identified. We also looked at variants that are annotated in ClinVar for *NOX4* in the enhancer/promoter regions (ClinVar dbSNP (155)) **rs139341533** chr11:89182666-89182666; **rs144387171** chr11:89182672-89182672; **rs140381222** chr11:89135668-89135668; **rs139363194** chr11:89133506-89133506; **rs111971665** chr11:89075367-89075367; and enhancer **rs115031759** chr11:89073269-89073269. https://genome.ucsc.edu/cgi-bin/hgTrackUi?hgsid=1778674868_1bW83Myn46sgAunAqaOFKKuLMP95&db=hg19&c=chr11&g=dbSnp155Composite. None of the variants were present in the five WGS PAM patients.

Of course, we acknowledge the limitation of our workflow, as indicated in our manuscript on (page 13, lines 297-300). ...‘In our genetic analysis, we did not find any other *NOX4* rare variants in the rest of the PAM cohort (n=57), but we cannot exclude that these patients may carry other variants located in intergenic or intragenic regions which would escape our analysis as most of the patients were sequenced only for exomes.’

Table R1: Mapping promotor/enhancer lesion of *NOX4* gene

Location in GRCh37/hg19	Promoter/enhancer regions
chr11:89,338,200-89,344,146	Upstream of the gene/enhancer
chr11:89,215,596-89,238,872	Promoter region
chr11:89,195,616-89,198,280	Intron in NOX4 /potentially an enhancer
chr11:89,174,111-89,177,684	Probably an enhancer, inside an intron
chr11:89,106,239-89,117,143	Strong signal, enhancer, in intron
chr11:89,088,458-89,089,859	Probably an enhancer
chr11:89,064,876-89,074,438	Enhancer, spans exons and introns
chr11:89,053,965-89,055,453	Downstream of NOX4 - probably enhancer

3) Figure 3. In overexpression system, transcript levels largely depend on the structures of each expression vectors. It is unclear how those artificial expression vectors represent phenomena found in their patients. Was there statistically significant difference in ROS levels between pcDNA3.1 and NOX4^{WT}?

Response: We appreciate your thoughtful consideration. The use of artificial expression vectors in cell line studies is a common and simplified *in vitro* technique, which is used for specific effects of single-gene investigation or molecular mechanisms under controlled conditions. While we acknowledge that these models may not fully replicate the complexity of the human body, they still provide valuable insights into *NOX4* function and ROS changes. The findings obtained from our *in vitro* models are essential to understand the functional role of the gene and its variants and provide hypotheses that need to be further validated in more complex systems. In our study this validation process included additional models such as *in vivo* - zebrafish model, measurement of ROS in whole blood samples from PAM patients, and primary cells- osteoclasts culture, which helped us to confirm the relevance of the observed effects.

Regarding the statistical analyses, two-tailed t-test indicate a statistically significant difference in mRNA expression (Figure R2A) and ROS levels between the pcDNA3.1 and NOX4^{WT} ($p=0.0222$) (Figure R2B and R2C), revealing that the presence of NOX4^{WT} has a substantial impact on ROS production compared to the control group.

Figure R2: Impact on HEK293 cell lines overexpressing pcDNA3.1 and NOX4^{wt}.

(A) Quantification of *NOX4* mRNA expression through qRT-PCR in HEK293 stable transfected cells expressing pcDNA3.1 (empty vector) and pcDNA3.1-*NOX4*^{wt} (wild-type). (B) ROS generation was elevated in cells overexpressing *NOX4*^{wt} compared to pcDNA3.1-transfected cells by EPR. (C) The H₂O₂ present in cells was assessed using the BioTracker Green H₂O₂ live cell dye in HEK293 stable cells followed by quantification of the fluorescence intensity. H₂O₂: Hydrogen Peroxide. Data are shown as mean \pm SEM based on the two-tailed T-test.

Figure 4

How the authors think of the effect of TNF-inhibitors on ROS production? When etanercept was predicted to increase ROS production, why adalimumab was considered to decrease that?

Response: We see that this point needs clarification. It has been shown in different studies that TNF- α induces the production of ROS (Kim *et al*, 2010; Zelova & Hosek, 2013). Therefore, TNF-inhibitors decrease ROS production. Adalimumab is a disease-modifying antirheumatic drug that inactivates TNF α . Both etanercept and adalimumab are TNF-inhibitors that target inflammation (we added text to clarify, see page 9, lines 213-215. ...'Several studies have shown that TNF- α induces the production of ROS (Kim *et al.*, 2010; Zelova & Hosek, 2013) and the medication could have affected the level of ROS.')

Figure 5

Osteoclast differentiation from peripheral blood is interesting but evaluated in only one patient.

Response: We understand the concern of the reviewer. We have incorporated data from one more patient PsA961 in this analysis, who is a Psoriatic Arthritis (PsA) patient carrying the rare NOX4 variant (p.Y512C). Interestingly, the osteoclasts from this patient show a similar pattern than PAM12, i.e., enhanced osteoclast differentiation and increased ROS (more specifically H₂O₂) activity. We have added the data into a new Figure 6.

As the reviewer may appreciate, it is extremely difficult to recruit more patients due to the rarity of PAM, the geographical dispersion of the patients living in different cities and countries, as well as the mobility issues associated with the disease. In addition, the ROS measurement by EPR and osteoclast differentiation require fresh blood samples and a minimum amount of the already limited CD14-positive cells, making it quite challenging.

Referee #2 (Comments on Novelty/Model System for Author):

Potentially, the findings improve the understanding of this severe form of psoriatic arthritis.

Referee #2 (Remarks for Author):

In this manuscript, Wang S. *et al.* studied a cohort of Nordic patients (Sweden, Denmark and Norway) with psoriatic arthritis mutilans (PAM). By using of next generation sequencing (whole exome sequencing and whole genome sequencing), they found rare variants in NOX4 gene, which are characterized in overexpression system and animal model (zebrafish) in terms of reactive oxygen species (ROS) production. The authors concluded that these variants cause a high ROS production. In addition, they found other variants in some genes related with ROS/NOX4 pathway. See my comments:

Response: We would like to express our sincere appreciation for your thoughtful and constructive comments on our manuscript.

-The model to find rare variants are missing in this article: autosomal recessive, autosomal dominant or X-recessive pattern? MAF?

-Familial segregation is also missing here. I assumed regarding the electropherogram that variants of patients are heterozygous.

Response: Thank you for the comment. We are sorry we were not clear on this subject. Our cohort consists of cases and controls, we do not have families. Therefore, it is not straightforward to discuss inheritance models.

Based on the data that we have, the disease seems to be autosomal dominant as the rare variants examined are potentially pathogenic in the carriers. The variants

were validated by several methods and are not present in healthy controls. Unfortunately, without material from parents, we are unable to confirm if the variants are *de novo* or inherited.

The variants are indeed heterozygous, we mention it in the legend of Figure 2b "*heterozygous variants in NOX4 are shown*". We also added the information in the results section (page 6, lines 129-130). ...'All variants found are heterozygous and have been confirmed by Sanger sequencing of genomic DNA.'

Regarding the rare variants the filtering is described in the flowchart of Figure 1, but it may be worth clarifying that the criteria in the first screening was to find *i*) very rare variants in the same gene in at least two patients (MAF<0.0001), *ii*) rare variants in different genes from the same pathway, or *iii*) rare homozygous or compound heterozygous loss-of-function variants in coding regions or variants that would affect gene-splicing. We explored the list of genes obtained in the first screening by thorough literature search, gene ontology (GO) and HPO annotations. *NOX4*, a gene involved in bone resorption stood out when the first five Swedish patients were sequenced (Figure 2). By finding another two rare variants in two PAM patients affecting important domains for gene function in the *NOX4* gene encouraged us to perform whole exome sequencing in the remaining 56 PAM patients.

-Expression of RNA was evaluated in HEK293 cell lines. **However, the authors do not show protein expression.**

Response: Thank you for pointing this aspect out. We started our attempt to understand the consequences of the rare variants found in *NOX4* patients by using HEK293 cells (Figure 3). These cells are easy to propagate, to transfect, and importantly the system has been used before in *Nox4* studies (Takac *et al*, 2011). Our primary focus was on understanding gene expression and the results obtained through RT-PCR consistently supported our findings.

We performed Western blots several times with extracts from the transiently transfected cells, the results showed the same trend as the results seen by RT-PCR and EPR but were not consistent enough to conclude that there was a significant change at the protein level in those cells. Western blot is a semi-quantitative method, influenced by various technical and biological factors. Therefore, we decided not to show the HEK293 western blots in our manuscript. However, to strengthen our observations, we have added a new figure (Figure 6D) demonstrating significantly elevated *NOX4* protein levels in osteoclasts from a patient carrying one of the rare variants (*NOX4*^{Y512C}) compared to a healthy control, providing additional insights into protein expression levels.

-The authors should specify that are the cells used for testing ROS production: whole blood, peripheral mononuclear cells, peripheral monocytes, neutrophils, osteoclast differentiation?

Response: Thank you for the comment. We have measured ROS levels by different methods, including Electron paramagnetic resonance (EPR), 2',7'-dichlorofluorescein diacetate (DCFH-DA) and a new H₂O₂ specific probe in the improved manuscript. To clarify see the table R2 below.

Table R2: The methodologies employed for each species of ROS measurement in our study.

	Stable transfected	Patient derived	Zebrafish	Patients whole
--	--------------------	-----------------	-----------	----------------

	HEK293 cells	osteoclasts ¹	model	blood samples
H ₂ O ₂ probe	+	+		
EPR	+			+
DCFH-DA	+	+	+	

¹Osteoclasts were differentiated from peripheral blood monocytes.

-Samples of PAM12 were used for the differentiation of osteoclasts and the evaluation of ROS production. What is here the connexion? She/he is mutated in NOX4?

Response: The patient PAM12 lives in our city and has visited our clinic. Despite extensive genome analysis, no candidate rare variants in *NOX4* nor in other genes that would account for the phenotype were identified (explained in page 9, lines 215-216). ...‘It should be noted here that we did not find any *NOX4* mutations in PAM12 and PAM 37, nor mutations in any other genes related to *NOX4*’.

It is possible that rare variants on *NOX4* or other genes affecting the same pathway might exist in inter- or intragenic regions and have eluded our analysis. Moreover, epigenetic mechanisms influencing gene expression cannot be ruled out.

In the improved manuscript we have added one additional patient for the osteoclast differentiation, the PsA patient (PsA961) carrying the rare variant *NOX4*^{Y512C}. The results from this patient are remarkably similar to the results obtained from the PAM12, reinforcing the consistency of our results in osteoclast differentiation (Figure 5B-C) and the generation of ROS (Figure 5D-E).

Referee #3 (Remarks for Author):

Wang et al have applied whole-genome sequencing (n=5) and whole exome sequencing (n=56) to search for potentially disease relevant gene polymorphisms in patients with Psoriatic arthritis mutilans (PAM); an infrequent and severe variant of psoriatic arthritis. They found rare variants in the NADPH oxidase 4 (*NOX4*) in four patients.

NOX4, a member of the *NOX*-family of enzymes that functions as the catalytic subunit of the NADPH oxidase complex, is known to be involved in the production of reactive oxygen species (ROS). The authors have tested in in vitro and in vivo models whether the identified variants are involved in enhancing ROS release. Specifically, they are presenting data from osteoclasts of 1 patient with PAM, from stably transfected cell lines, from direct measurements of superoxide in patient blood samples and from zebra fish models. They conclude that *NOX4* is a susceptibility gene in PAM and contributes to disease through production of high levels of superoxide, specifically in osteoclast.

Comments:

1. PAM is a disabling subtype of psoriatic arthritis and genetic risk factors have not yet been identified. Therefore, the current study is timely and potentially impactful.

2. The authors have available a cohort of n=61 PAM patients that have been clinically well phenotyped. This is a valuable resource.

Response: We thank the reviewer for the positive observations and feedback and considered the important comments, your insights and suggestions have been invaluable in enhancing the quality and clarity of our work.

3. While it is straightforward to speculate that a polymorphism in a NOX gene is involved in ROS production, the mechanistic data presented in the current manuscript remain unconvincing.

Response: In response to the concern raised about the mechanistic data, we have expanded our analysis and included additional experiments to provide a more robust mechanistic insight into the involvement of *NOX4* variants. Please see in the point-by-point answers below.

4. As a general rule, the study cohorts for the mechanistic experiments are very small, often consisting of only one patient.

Response: We understand the concern of the reviewer and it is a major limitation in our study as mentioned in page 16, lines 367-369. ...'A limitation of the present study is the lack of access to fresh blood samples from the majority of PAM patients, needed for measurement of ROS by EPR and for osteoclast differentiation from CD14+ monocytes. Our study would have benefited from exploring ROS levels in more patients.'

We answered a similar question from reviewer 1. Our answer is that *"..it is extremely difficult to recruit more patients due to the rarity of PAM, the geographical dispersion of the patients' living in different cities and countries, as well as the mobility issues associated with the disease. In addition, the ROS measurement by EPR and osteoclast differentiation require fresh blood samples and a minimum amount of the already limited CD14-positive cells, making it quite challenging"*

However, we have added one more patient carrier of the *NOX4* rare variant (*NOX4*^{Y512C}), one more PsA patient (without the *NOX4* rare variants) and one more control to our analyses in osteoclast differentiation experiments. See New Figure 6 and Figure 5 and Figure EV2.

5. *Nox4* is unusual as it releases hydrogen peroxide (H₂O₂) in contrast to *Nox1*-*Nox3* and *Nox5*, which release superoxide (Nisimoto et al, Biochemistry 2014). Curiously, the authors present several data sets showing that *NOX4* generates superoxide and the title of the manuscript implicates *NOX4* in inducing high superoxide levels. This is a critical claim of the manuscript and requires clarification.

Response: Thank you for bringing this up, it is indeed an important point. In the cited paper (Nisimoto, 2014) (Nisimoto *et al*, 2014), approximately 90% of the electron flux through *Nox4* produces H₂O₂ and 10% forms superoxide. We apologize that we did not make this clearer in first instance.

Regarding the measurements with EPR, EPR probes can rapidly react with all types of oxygen-centered free radicals including superoxide, although superoxide measurement is the main application of EPR probe. The reviewer is right that it may be better to use "ROS levels" instead of superoxide when discussing the results by EPR. We understand the confusion, and to be more correct we have changed through the text, including the title of our study to: "Rare coding variants in *NOX4* link high ROS levels to psoriatic arthritis mutilans" (marked in red).

In 2022 a guideline to assess ROS in cells and in vivo was published in Nature Metabolism (Murphy *et al*, 2022). We followed the guidelines on how to best assess specific ROS and acquired a commercially available live cell dye probe that detects H₂O₂. The figure below (Figure R3) shows the specificity of the probe in detecting H₂O₂ compared to other reactive species. The probe has been used recently in (Eisenbeis *et al*, 2023).

https://www.merckmillipore.com/SE/en/product/BioTracker-Green-H2O2-Live-Cell-Dye,MM_NF-SCT039?ReferrerURL=https%3A%2F%2Fwww.google.com%2F

Quality Control

Purity: $\geq 90\%$ confirmed by LC.

Figure R3: Reaction of BioTracker™ Green H₂O₂ dye with various reactive oxygen species. Source: Sigma (probe catalog number SCT039).

In the new version of our manuscript we have added new experiments in which we used the probe mentioned above: BioTracker Green H₂O₂ Live Cell Dye (SCT039, sigma) (Abo *et al*, 2011), specifically detecting H₂O₂. We used the probe in both stable transfected HEK293 cell lines (New Fig 3C-D) and in patient derived osteoclasts (New Fig 6B-C). Our revised experiments and the utilization of a highly specific H₂O₂ probe confirmed the increase of H₂O₂ in our samples, aligning with the existing understanding of NOX4's dual ROS production capabilities. We hope this clarification and the added experiments addressed your concerns adequately.

6. Not only does Nox4 emit a different pattern of ROS, but there are fundamental differences between NOX4 and the other members of the NOX family. Specifically, its subcellular localizations, tissue distribution and influence over signaling pathways is different from the other Nox enzymes. Nox4 is recognized for its role in oxygen sensing, vasomotor control, cellular proliferation, differentiation, migration, apoptosis, senescence, fibrosis, and angiogenesis.

The current manuscript is focused exclusively on the expression of NOX4 in osteoclastic cells, a subtype of myeloid cells. This appears to be a missed opportunity and it is not clear how the authors arrived at this focus. Here, an unbiased approach would make more sense.

Response: NOX4 is a remarkable molecule with multiple roles, and despite extensive studies there are still many aspects that need further studies, specifically its role in disease. We understand that from the reviewer's perspective, we are missing an opportunity to study other aspects of NOX4 but our focus is the study of PAM. This is the first paper linking PAM to a candidate gene, and we think that our study is opening the door to further studies on NOX4 biology as well as deeper clinical aspects of PAM.

Our genetic approach was unbiased, we did not have any candidate gene *a priori* to test, and we described the approach answering a question from reviewer 1, see below:

Regarding the rare variants the filtering is described in the flowchart of Figure 1, but it may be worth clarifying that the criteria that we had in our first screening were to find **i)** very rare variants in the same gene in at least two patients (MAF<0.0001), **ii)** rare variants in different genes from the same pathway, or **iii)** rare homozygous or compound heterozygous loss-of-function variants in coding regions or variants that would affect gene-splicing. We explored the list of genes obtained in the first screening by thorough literature search, gene ontology (GO) and HPO annotations.

Osteoclasts are the most important cells in PAM, they resorb the bone and play a crucial role in bone homeostasis. The PAM patients suffer from osteolysis and bone resorption in fingers and toes. NOX4 has shown to be particularly important in bone resorption, the main characteristic of PAM, therefore we look specifically to osteoclasts in patients (we mention the importance of osteoclasts in results section, page 10, lines 233 and page 15, lines 347-354). ...'NOX4 is induced during osteoclast differentiation, the cells responsible for bone resorption.' and ...'In addition, NOX4 is important for osteoclast differentiation. Osteoclasts are multinucleated cells derived from the monocyte macrophage lineage, important for bone remodeling. They resorb the bone and its hyperactivated function is implicated in diseases such as osteoporosis, periprosthetic osteolysis, Paget's bone disease, and rheumatoid arthritis (Bi et al, 2017). Here, we evaluated osteoclast differentiation and generation of ROS in one PAM patient (PAM12) and one PsA patient carrying a NOX4 rare variant (PsA961 carrier of NOX4Y512C) and found that the patients showed a similar pattern compared to a PsA individual without NOX4 rare variants and two healthy controls.'

Figure for reviewers removed, showing NOX4 upregulation and NOX2/NOX1 downregulation in mature osteoclasts.

Taken from <https://www.sciencedirect.com/science/article/pii/S0891584918315028>

7. The Human Protein Atlas provides data about the expression of NOX4 in different cell types. By far the highest expression of NOX4 occurs in proximal tubular cells in the kidney. To a much lower degree is NOX4 found in fibroblasts and smooth muscle cells in different organs. It is explicitly noted that immune cells, including the myeloid cell types, are low for NOX4 expression. Here, it may be important to consider how NOX4+ proximal tubular cells in the kidney could have a potential role in PAM.

Response: *NOX4* expression was originally described in the kidneys due to its abundance in tubular cells (Shiose *et al*, 2001). Subsequent research has revealed that *NOX4* is constitutively active and has roles in multiple tissues. While in some patients with psoriasis and psoriatic arthritis the renal function is impaired (Khan *et al*, 2017), in PAM patients no kidney malfunction is observed.

As described in the previous response, the osteoclasts are the most important cells to study the PAM patients' phenotype, that is characterized by osteolysis and bone resorption in fingers and toes. Unfortunately, bone cells are not depicted in the human protein atlas.

We recognize the importance of considering the broader implications of *NOX4* biology, and future studies exploring its potential involvement in kidney cells could contribute valuable insights beyond the scope of our current investigation.

8. NOX4 is overexpressed in some tumors and has been implicated in regulating oncogenic metabolic adaptations. This is another important aspect of NOX4 biology that has not been considered in the current manuscript.

Response: As PAM is a rare disease there is difficult to know if the PAM patients have increased risk of cancer. We focus our study in the bone aspects of PAM. As our primary objective is to investigate the genetic basis of PAM and its impact on osteoclasts, our study does not extend to exploring the potential association between NOX4 and tumorigenesis in PAM patients.

9. Figure 3: The authors are showing highly elevated NOX4 RNA expression in HEK293 cell lines transfected with a vector carrying wt or variant NOX4. Did they control for how many copies of the vector each of the cell lines contains?

Response: Thank you for your insightful question regarding the control for vector copy number in our study. During the generation of our stable cell lines, single clones were plated in 96-well plates. We minimized the potential transfer of multiple clones and, subsequently, reduced variability within the transfected cell lines. To quantify the vector copy number in our cell lines, we used quantitative PCR (qPCR). Firstly, we designed primers targeting the F1ori region, specifically binding to the pcDNA3.1-hNox4 plasmid (#69352, addgene) (Table R3). Later, we compared the Ct values after extracting genomic DNA from the stable HEK293 transfected cell lines. No differences were observed between the vector copy numbers among the *NOX4* rare variants when compared to *NOX4^{wt}* (Figure R5). The relative copy numbers were not significantly different between all cell line, suggesting a comparable level of vector integration.

Table R3: The primers used in quantitative PCR

Primer Name	Sequence (5' to 3')	Binding Sites (bp)	Length (bases)
F1ori-F	GTGGACTCTTGTTCCAAACTGG	3275 - 3296	22
F1ori-R	AGGGAAGAAAGCGAAAGGAG	3065 - 3084	20

Figure R5: Relative vector copy numbers in HEK 293. The figure illustrates the relative vector copy numbers of F1ori, analyzed through quantitative PCR (qPCR) in HEK293 stable transfected cells expressing pcDNA3.1 (empty vector), $NOX4^{wt}$ (wild-type), $NOX4^{Y512IfsX20}$, $NOX4^{Y512C}$, and $NOX4^{V369F}$. *GAPDH* expression was used as the loading control. The data represent the mean values from three independent experiments, and statistical analysis was conducted using one-way ANOVA.

10. Figure 3: The expression of NOX4 RNA appears to be massively higher if HEK293 cells are transfected with vectors containing NOX4(Y512IfsX20), NOX4(Y512C), and NOX4(V369F). If correct, this is an interesting observation. Do the authors know what the underlying mechanism is?

The authors state that the polymorphisms affect the NADPH binding domain of the enzyme (Line 309). Why would that lead to a change in RNA abundance? This is difficult to understand.

Response: Thank you for the comment. This is an observation; our experiments were repeated several times with similar results. One explanation could be that the rare variants could affect the protein folding, prompting compensatory elevation in gene expression. We explored this hypothesis by *in silico* analysis using the DUET tool (a server for predicting effects of mutations on protein stability using an integrated computational approach) (Pires *et al*, 2014), which predicts protein stability. The data suggests that the variants affecting the NADPH binding and FAD binding domains could indeed affect protein stability, see Figure R6.

We plan to explore further the underlying mechanisms in future investigations. However, our current focus is on the understanding the effects of the variants regarding PAM disease.

DUET - Protein Stability Change Upon Mutation

DUET - Protein Stability Change Upon Mutation

Figure R6: Result of DUET prediction for NOX4(Y512C) and NOX4(V369F).

11. Figure 3B. The authors show higher superoxide expression in the cell lines transfected with NOX4 variants. NOX4 releases hydrogen peroxide (H_2O_2) in contrast to Nox1-Nox3 and Nox5, which release superoxide (Nisimoto et al, Biochemistry). Here, it would be critical to show production of hydrogen peroxide.

Response: Thanks for your valuable suggestions. We acknowledge the importance of demonstrating hydrogen peroxide (H_2O_2) production in our study. Approximately 90% of the electron flux through Nox4 results in the generation of hydrogen peroxide (H_2O_2), with the remaining 10% resulting in the formation of superoxide (Nisimoto *et al.*, 2014). To specifically detect H_2O_2 , we performed additional experiments using a fluorescent probe, which reacts with H_2O_2 but does not react with other ROS species such as hydroxyl radical ($\cdot OH$), superoxide (O_2^-), hypochlorous acid (HOCl), singlet oxygen (1O_2), and nitric oxide (NO) as illustrated in the reply to a previous question (Figure R3). Our results demonstrated a significant increase in fluorescence intensity in cells expressing $NOX4^{Y512I/SX20}$, $NOX4^{Y512C}$, and $NOX4^{V369F}$ compared with pcDNA3.1 (empty vector) and $NOX4^{wt}$ (wild-type) (new

Figure 3C) and (text in page 9, Lines 197-200). ...Furthermore, we assessed overall ROS in HEK293 cells through DCFH-DA stainings (Figure EV1) and employed a specific probe to evaluate H₂O₂ levels (Figure 3C). We observed higher ROS and H₂O₂ levels in all cells expressing the rare variants compared to those overexpressing NOX4^{wt}.

This observed trend in fluorescence intensity aligns with the results obtained for ROS production.

12. Line 426-428: "The results suggest that the variant NOX4 (Y512C) affecting the NADPH binding domain is responsible for the elevated superoxide production."

Are the authors proposing that the rare variants of NOX4 change emission of the ROS pattern? To support that claim, they would need to show precise measurements of H₂O₂ and of superoxide (ideally in a membrane-free system).

Response: Thank you for your insightful comment. The statement in lines 426-428 proposes the expression of superoxide in PsA961 (a carrier of NOX4^{Y512C}) is higher compared with healthy control, psoriasis, and psoriasis arthritis without NOX4 variants by the electron paramagnetic resonance (EPR), which is a highly sensitive and unique method that allows the direct detection of ROS (Figure 3B). The results support the hypothesis that NOX4^{Y512C} alters the emission pattern of superoxide by affecting the NADPH binding domain. Using a new probe described in the previous question we also detect that the levels of H₂O₂ are significant higher (Figure 3C).

In addition, in a new experiment and to further elucidate the implications of NOX4^{Y512C} we differentiated osteoclasts from PsA961-derived peripheral blood mononuclear cells. Notably, induced osteoclasts exhibited higher levels of H₂O₂ compared to uninduced osteoclasts. This observation aligns with our principal hypothesis, suggesting that the NOX4^{Y512C} variant contributes to an altered ROS pattern, affecting both superoxide and hydrogen peroxide production.

The higher levels of H₂O₂ are seen in all the systems (stable cells, Figure 3; and in zebrafish Figure 6) and is examined not only in the NOX4^{Y512C} rare variant but also by the other rare variants found NOX4^{Y512fsX20}, and NOX4^{V369F}.

13. Line 429-430: "It should be noted here that we did not find any NOX4 mutations in PAM12 and PAM37, nor mutations in any other genes related to NOX4."

Nevertheless, the authors chose PAM12 and PAM37 to examine peripheral blood ROS measurements. PAM12 appears to produce high levels and PAM37 appears to produce levels indistinguishable from controls.

- The authors should clarify what "peripheral blood" is. Is this serum? Plasma? Does it contain cells?

Response: We appreciate your attention to the details of our study. Peripheral blood refers to whole blood collected from individuals. This includes all blood components, such as red blood cells, white blood cells, platelets, and plasma.

- It is unclear why they examined patients that do not carry the variant.

Response: Thank you for the question. We understand that it is a concern that the examined patients do not carry the rare variants.

Reviewer 1 pointed that out a similar question, see our reply below:

We understand the concern of the reviewer and we have added one more patient in this analysis, patient PsA961 who is a PsA patient and carries the rare *NOX4* variant (p.Y512C). Interestingly, the osteoclasts from the patient show a similar pattern than PAM12, i.e., enhanced osteoclast differentiation and increased ROS activity.

We have added the data into a new Figure 6 and Figure EV2.

It is difficult to recruit more patients, PAM is a rare disease and the patients included in our Nordic cohort live in different cities, in different countries are often elderly and have difficult to travel due to their condition. In addition, to differentiate osteoclasts is not an easy procedure, fresh blood samples are required, the number of cells after CD14+ selection is limited, and the cells cannot be propagated.

In addition, PAM12 lives within our city and has made multiple visits to our clinic. We conducted a thorough examination of his genome, yet no rare variants that would account for the phenotype were identified. It is conceivable that potential rare variants within *NOX4* or other genes affecting the same pathway might exist in inter- or intragenic regions and have eluded our analysis. Additionally, epigenetic mechanisms influencing gene expression cannot be ruled out.

PAM12, being a focal point in this underwent comprehensive examination. In the new version of the manuscript, we have added one additional patient and the results are remarkably similar to the results obtained from the PAM12 regarding osteoclast differentiation (Figure 5B) and the generation of ROS (Figure 5D).

- To examine a single patient is insufficient to draw any conclusions. Here, they need to examine patient cohorts of sufficient size and include patients with other inflammatory diseases as controls. High ROS production has been implicated in a variety of inflammatory diseases. To support their claim, the authors need to link the high ROS production to *NOX4*.

Response: Thank you for the comment. We acknowledge the comments from reviewer 3 prompting us to use more specific probes. The study has been improved and in the new version we added one probe to detect H₂O₂. We also added one patient in our study and performed the experiments related to the osteoclast differentiation.

14. In some experiments, the authors have used DCFH-DA for "ROS quantification". Quantifying specific subtypes of ROS is challenging, particularly for H₂O₂ (see recently developed Guidelines; Murphy et al, Nature Metabolism, 2022). DCFH does not directly react with H₂O₂ to form the fluorescent product, DCF. Therefore, DCF fluorescence cannot be used as a direct measure of H₂O₂. Experiments utilizing DCFH-DA should be interpreted with caution.

Response: Thank you for the valuable comment.

We agree that DCFH-DA is commonly used as a general indicator of cellular oxidative stress, reacting with a variety of ROS, including superoxide anions, hydroxyl radicals, and peroxynitrite, among others.

We appreciate your emphasis on the challenges associated with quantifying specific ROS subtypes, especially H₂O₂, as highlighted in the recent guidelines by Murphy et al. (Nature Metabolism, 2022). While DCFH-DA may not directly react with H₂O₂ to form the fluorescent product DCF, it remains a useful fluorescent probe capable of detecting intracellular H₂O₂. Its broad reactivity profile allows for a comprehensive assessment of overall oxidative stress dynamics. We concur that experiments utilizing DCFH-DA should be interpreted with caution.

In our experimental design, we have implemented additional assays to enhance the reliability of our oxidative stress measurements.

References

- Abo M, Urano Y, Hanaoka K, Terai T, Komatsu T, Nagano T (2011) Development of a highly sensitive fluorescence probe for hydrogen peroxide. *J Am Chem Soc* 133: 10629-10637
- Busch M, Schwindt H, Brandt A, Beier M, Gorltd N, Romaniuk P, Toska E, Roberts S, Royer HD, Royer-Pokora B (2014) Classification of a frameshift/extended and a stop mutation in WT1 as gain-of-function mutations that activate cell cycle genes and promote Wilms tumour cell proliferation. *Hum Mol Genet* 23: 3958-3974
- Eisenbeis VB, Qiu D, Gorka O, Strotmann L, Liu G, Prucker I, Su XB, Wilson MSC, Ritter K, Loenarz C *et al* (2023) beta-lapachone regulates mammalian inositol pyrophosphate levels in an NQO1- and oxygen-dependent manner. *Proc Natl Acad Sci U S A* 120: e2306868120
- Ho MS, Tsang KY, Lo RL, Susic M, Makitie O, Chan TW, Ng VC, Sillence DO, Boot-Handford RP, Gibson G *et al* (2007) COL10A1 nonsense and frame-shift mutations have a gain-of-function effect on the growth plate in human and mouse metaphyseal chondrodysplasia type Schmid. *Hum Mol Genet* 16: 1201-1215
- Khan A, Haider I, Ayub M, Humayun M (2017) Psoriatic Arthritis Is an Indicator of Significant Renal Damage in Patients with Psoriasis: An Observational and Epidemiological Study. *Int J Inflamm* 2017: 5217687
- Kim JJ, Lee SB, Park JK, Yoo YD (2010) TNF-alpha-induced ROS production triggering apoptosis is directly linked to Romo1 and Bcl-X(L). *Cell Death Differ* 17: 1420-1434
- Mair B, Konopka T, Kerzendorfer C, Sleiman K, Salic S, Serra V, Muellner MK, Theodorou V, Nijman SM (2016) Gain- and Loss-of-Function Mutations in the Breast Cancer Gene GATA3 Result in Differential Drug Sensitivity. *PLoS Genet* 12: e1006279
- Murphy MP, Bayir H, Belousov V, Chang CJ, Davies KJA, Davies MJ, Dick TP, Finkel T, Forman HJ, Janssen-Heininger Y *et al* (2022) Guidelines for measuring reactive oxygen species and oxidative damage in cells and in vivo. *Nat Metab* 4: 651-662
- Nisimoto Y, Diebold BA, Cosentino-Gomes D, Lambeth JD (2014) Nox4: a hydrogen peroxide-generating oxygen sensor. *Biochemistry* 53: 5111-5120
- Pires DE, Ascher DB, Blundell TL (2014) DUET: a server for predicting effects of mutations on protein stability using an integrated computational approach. *Nucleic Acids Res* 42: W314-319
- Schroder K (2019) NADPH oxidases in bone homeostasis and osteoporosis. *Free Radic Biol Med* 132: 67-72
- Shiose A, Kuroda J, Tsuruya K, Hirai M, Hirakata H, Naito S, Hattori M, Sakaki Y, Sumimoto H (2001) A novel superoxide-producing NAD(P)H oxidase in kidney. *J Biol Chem* 276: 1417-1423
- Takac I, Schroder K, Zhang L, Lardy B, Anilkumar N, Lambeth JD, Shah AM, Morel F, Brandes RP (2011) The E-loop is involved in hydrogen peroxide formation by the NADPH oxidase Nox4. *J Biol Chem* 286: 13304-13313
- Zelova H, Hosek J (2013) TNF-alpha signalling and inflammation: interactions between old acquaintances. *Inflamm Res* 62: 641-651

11th Jan 2024

Dear Dr. Tapia-Paez,

Thank you for submitting your revised manuscript, and please accept my apologies for the delay in getting back to you in this busy time of the year. We have now received the reports from referees #1 and #2 who re-reviewed your manuscript. Referee #1 also evaluated your responses to referee #3's concerns. As you will see below, they are supportive of publication pending minor revisions, and I will therefore be able to accept your manuscript once the following points will be addressed:

1/ Referees' comments:

Please address the remaining concerns mentioned by referee #1.

2/ Manuscript text:

- Please remove the red text, and only keep in track changes mode any new modification.

- Materials and Methods:

o Cells: please provide culture conditions and indicate whether the cells were authenticated and tested for mycoplasma contamination.

o Statistics: please include a statement on blinding, randomization, and inclusion/exclusion criteria.

o Please correct the checklist accordingly.

- Data availability: Please remove "All data supporting the conclusions in the article are presented in the main text or supplementary data files. Additional data are available from the corresponding author upon request."

Primary datasets produced in this study need to be deposited in an appropriate public database, and the accession numbers and database listed under 'Data Availability'. In case you have no data that requires deposition in a public database, please state so in this section ('This study includes no data deposited in external repositories.'). Note that the Data Availability Section is restricted to new primary data that are part of this study.

- Please rename "Conflict of interest" to "Disclosure statement and competing interests".

- Data citation:

1. Please note that the data citation does not refer to deposited experimental data, but refers to journal article.

2. Please note that the URLs for Data ref: (Ameur et al, 2017) and Data ref: (Marety et al, 2017) data citations are not provided.

3/ Figures and Appendix:

- Please carefully check the composition of your Figure 6 panel B: we noted a re-use of image between Day 7 Psa961 / M+R and day 9 C17 M , which doesn't match the source data provided. Please correct.

- Appendix: tables should be renamed "Appendix Table S1" etc. The second appendix file with structural variants should either be added to the main appendix file as "Appendix Table S8" or uploaded as an excel file labeled "Dataset EV1", with a short legend added

- Figure legends:

1. Please note that the data information for figure panels 3a-c is incorrectly mentioned as 3c in the legend of figure 3. This needs to be rectified.

2. Please note that the gels and blot image of figure 5d is not described in the corresponding legend. This needs to be rectified.

3. Please note that the box plots need to be defined in terms of minima, maxima, centre, bounds of box and whiskers, and percentile in the legends of figures 4a-c.

4. Please note that n=2 in figures 4a-c.

5. Please note that the scale bar needs to be defined for figure EV 2a.

4/ Checklist:

Please complete the following sections: Cell authentication/contamination - Experimental study design and statistics - Data Availability

5/ Synopsis:

Please upload your synopsis text as a word document.

6/ As part of the EMBO Publications transparent editorial process initiative (see our Editorial at

<http://embomolmed.embopress.org/content/2/9/329>), EMBO Molecular Medicine will publish online a Review Process File (RPF) to accompany accepted manuscripts.

This file will be published in conjunction with your paper and will include the anonymous referee reports, your point-by-point response and all pertinent correspondence relating to the manuscript. Let us know whether you agree with the publication of the RPF and as here, if you want to remove or not any figures from it prior to publication.

I look forward to receiving your revised manuscript at your earliest convenience

Yours sincerely,

Lise Roth

***** Reviewer's comments *****

Referee #1 (Remarks for Author):

The authors almost successfully addressed to the queries raised by this reviewer and also by Reviewer #3. In Figure 3, protein levels of NOX4 by Western blot (or other method) would be necessary to at least partially address the question raised from these reviewers, in order to connect mRNA levels and ROS production comparing those variants. In Figure 6A, does "PsA new" correspond to PsA77?

Referee #2 (Remarks for Author):

The authors have addressed my comments.

***** Reviewer's comments *****

Referee #1 (Remarks for Author):

The authors almost successfully addressed to the queries raised by this reviewer and also by Reviewer #3.

We appreciate the Review's acknowledgment of our efforts to improve the manuscript and we are pleased to address the remaining concerns to further enhance the quality of the manuscript.

In Figure 3, protein levels of NOX4 by Western blot (or other method) would be necessary to at least partially address the question raised from these reviewers, in order to connect mRNA levels and ROS production comparing those variants.

We answered a similar question in the previous round to reviewer 2. See below:

Q Expression of RNA was evaluated in HEK293 cell lines. **However, the authors do not show protein expression.** “..Thank you for pointing this aspect out. We started our attempt

to understand the consequences of the rare variants found in NOX4 patients by using HEK293 cells (Figure 3). These cells are easy to propagate, to transfect, and importantly the system has been used before in Nox4 studies (Takac et al, 2011). Our primary focus was on understanding gene expression and the results obtained through RT-PCR consistently supported our findings.

We performed Western blots several times with extracts from the transiently transfected cells, the results showed the same trend as the results seen by RT-PCR and EPR but were not consistent enough to conclude that there was a significant change at the protein level in those cells. Western blot is a semi-quantitative method, influenced by various technical and biological factors. Therefore, we decided not to show the HEK293 western blots in our manuscript. However, to strengthen our observations, we have added a new figure (Figure 6D) demonstrating significantly elevated NOX4 protein levels in osteoclasts from a patient carrying one of the rare variants (NOX4^{Y512C}) compared to a healthy control, providing additional insights into protein expression levels.”

We add two figures below to show the NOX4 protein levels in HEK293 cells. Through Western blot analysis, we observed elevated NOX4 expression in transfected HEK293 cells expressing NOX4^{Y512fsX20}, NOX4^{Y512C}, and NOX4^{V369F} in comparison to NOX4^{wt} (wild-type). These findings contribute valuable insights, establishing a link between mRNA levels and ROS production across the different variants. We believe these additions significantly

enhance the comprehensiveness of our study. Thank you for the suggestion, and we hope this strengthens the overall scientific rigor of the manuscript.

Figure R7. **NOX4 protein expression in HEK293 cells.**

A. Western blot analysis of NOX4 in HEK293 stable cells expressing various constructs: *NOX4^{wt}* (wild-type), *NOX4^{Y512fsX20}*, *NOX4^{Y512C}*, and *NOX4^{V369F}*. GAPDH was used as a loading control. Quantification to the left. B. Replication of NOX4 protein expression analysis in HEK293 cells.

In Figure 6A, does "PsA new" correspond to PsA77?

We thank referee's careful observation, the label "PsA new" has been corrected into PsA77 in Figure 6A.

Referee #2 (Remarks for Author):

The authors have addressed my comments.

We are happy to see Referee's satisfaction with our addressing and the revised manuscript.

Takac I, Schroder K, Zhang L, Lardy B, Anilkumar N, Lambeth JD, Shah AM, Morel F, Brandes RP (2011) The E-loop is involved in hydrogen peroxide formation by the NADPH oxidase Nox4. *J Biol Chem* 286: 13304-13313

26th Jan 2024

Dear Dr. Tapia-Paez,

I am pleased to inform you that your manuscript is accepted for publication and is now being sent to our publisher to be included in the next available issue of EMBO Molecular Medicine!

With kind regards,

Lise Roth
